# Multiple tumor suppressors regulate a HIF-dependent negative feedback loop via ISGF3 in human clear cell renal cancer

Lili Liao[1,2†], Zongzhi Z Liu[2†], Lauren Langbein[1†], Weijia Cai[1†], Eun-Ah Cho[1,3†], Jie Na[4], Xiaohua Niu[5], Wei Jiang[1], Zhijiu Zhong[6], Wesley L Cai[2], Geetha Jagannathan[1], Essel Dulaimi[3], Joseph R Testa[3], Robert G Uzzo[3], Yuxin Wang[7], George R Stark[7], Jianxin Sun[8], Stephen Peiper[1], Yaomin Xu[9,10*], Qin Yan[2*], Haifeng Yang[1*]

[1]Department of Pathology, Anatomy and Cell Biology, Thomas Jefferson University, Pennsylvania, United States; [2]Department of Pathology, Yale University, Connecticut, United States; [3]Fox Chase Cancer Center, Pennsylvania, United States; [4]Department of Health Sciences Research, Mayo Clinic, Minnesota, United States; [5]Department of Gastrointestinal Surgery, The Sixth Affiliated Hospital of Guangzhou Medical University, Guangzhou, China; [6]Sidney Kimmel Cancer Center, Thomas Jefferson University, Pennsylvania, United States; [7]Department of Cancer Biology, Lerner Research Institute, Cleveland Clinic, Ohio, United States; [8]Department of Medicine, Thomas Jefferson University, Pennsylvania, United States; [9]Department of Biostatistics, Vanderbilt University Medical Center, Tennessee, United States; [10]Department of Biomedical Informatics, Vanderbilt University Medical Center, Tennessee, United States

*For correspondence:
yaomin.xu@vanderbilt.edu (YX);
qin.yan@yale.edu (QY);
Haifeng.yang@jefferson.edu (HY)

†These authors contributed equally to this work

Competing interests: The authors declare that no competing interests exist.

**Abstract** Whereas *VHL* inactivation is a primary event in clear cell renal cell carcinoma (ccRCC), the precise mechanism(s) of how this interacts with the secondary mutations in tumor suppressor genes, including *PBRM1*, *KDM5C/JARID1C*, *SETD2*, and/or *BAP1*, remains unclear. Gene expression analyses reveal that VHL, PBRM1, or KDM5C share a common regulation of interferon response expression signature. Loss of HIF2$\alpha$, PBRM1, or KDM5C in *VHL-/-*cells reduces the expression of interferon stimulated gene factor 3 (ISGF3), a transcription factor that regulates the interferon signature. Moreover, loss of SETD2 or BAP1 also reduces the ISGF3 level. Finally, ISGF3 is strongly tumor-suppressive in a xenograft model as its loss significantly enhances tumor growth. Conversely, reactivation of ISGF3 retards tumor growth by PBRM1-deficient ccRCC cells. Thus after *VHL* inactivation, HIF induces ISGF3, which is reversed by the loss of secondary tumor suppressors, suggesting that this is a key negative feedback loop in ccRCC.

## Introduction

Kidney cancer is among the top ten cancers, both in incidence and mortality in both men and women. Inactivation of the *VHL* tumor suppressor gene is a causal event in the pathogenesis of clear cell Renal Cell Carcinoma (ccRCC), the most frequent subtype of kidney cancer. Approximately 70 – 80% of ccRCC are sporadic tumors that harbor biallelic inactivation of *VHL* (*Linehan et al., 2004*). In the rare disease of hereditary kidney cancer, germline *VHL* mutation leads to early-onset bilateral kidney tumors. Biochemically, the protein product of the *VHL* tumor suppressor gene pVHL acts as the substrate recognition module of an E3 ubiquitin ligase complex. This complex targets the α sub-units of the heterodimeric transcription factor Hypoxia-Inducible Factor (HIF) for poly-ubiquitylation

and proteasomal degradation (*Zhang and Yang, 2012*). When HIFα is hydroxylated on either of two prolyl residues by members of the EglN family (also called PHDs or HPHs) under normal oxygen tension, it is recognized by pVHL. Without pVHL, HIFα protein accumulates and activates the hypoxia response transcriptional program. This constitutively active HIF subsequently drives ccRCC tumorigenesis and tumor growth (*Kaelin, 2005*). Interestingly, HIF targets include both tumor-promoting and tumor-suppressive genes, but its overall activity is potently oncogenic (*Zhang et al., 2013*).

Restoration of pVHL in *VHL-/-* ccRCC cells suppresses their ability to form tumors in immune-compromised mice, while stabilization of HIF2α overrides the effect of pVHL (*Kondo et al., 2002*). Conversely, HIF2α suppression in *VHL-/-* ccRCC cells cripples their ability to form tumors (*Zimmer et al., 2004*). The current standard of care for metastatic ccRCC focuses on inhibiting the VEGF receptor, yet drug resistance eventually develops in most cases. Thus it is urgent to identify new drug targets that operate in a high percentage of RCC tumors, with the expectation that the dual actions on VEGFR and a new target might prevent drug resistance.

Most types of solid tumors harbor multiple, sometimes dozens of mutations, in cancer genes to establish the hallmarks of cancer (*Hanahan and Weinberg, 2011*; *Vogelstein et al., 2013*). In addition to *VHL*, several frequently mutated genes were identified in sporadic ccRCC. Varela et al. reported that 41% of ccRCC tumors had inactivating mutations in the *PBRM1* gene. PBRM1 is a specificity subunit of the SWI/SNF chromatin-remodeling complex (*Varela et al., 2011*). The high mutation rate of *PBRM1* in ccRCC has been confirmed by multiple studies, together with mutations in other genes such as *BAP1*, *SETD2*, *KDM5C/JARID1C*, *PTEN* and *UTX* (*Dalgliesh et al., 2010*; *Guo et al., 2012*; *Peña-Llopis et al., 2012*; *Cancer Genome Atlas Research Network, 2013*; *Sato et al., 2013*). However, the mutation rates of the other genes are much lower than that of *PBRM1* (*Liao et al., 2015*).

Multiple lines of evidence suggest that *PBRM1* is a key tumor suppressor. Its mutations are predominantly inactivating in both alleles. PBRM1 suppression causes changes in pathways regulating chromosome instability and cell proliferation (*Varela et al., 2011*). Like *VHL* mutations, many *PBRM1* mutations occur early in tumorigenesis, unlike the other secondary mutations (*Gerlinger et al., 2012*). Recently a *PBRM1* germline mutation was reported to predispose patients to ccRCC (*Benusiglio et al., 2015*). PBRM1 was also found to amplify a HIF signature (*Gao et al., 2017*) and genetic ablation of both *Vhl* and *Pbrm1* in mouse kidneys leads to ccRCC while single loss fails to do so (*Nargund et al., 2017*; *Gu et al., 2017*).

KDM5C/JARID1C is a histone demethylase that removes methyl groups from tri-methylated lysine four on histone H3 (H3K4me3). H3K4me3 is a histone mark that is tightly linked to actively transcribed genes (*Barski et al., 2007*). *KDM5C* mutations occur in 3–7% of ccRCC tumors (*Varela et al., 2011*; *Dalgliesh et al., 2010*; *Cancer Genome Atlas Research Network, 2013*; *Sato et al., 2013*). Its mutations are mostly subclonal and happen later during tumor development (*Gerlinger et al., 2012*; *Gerlinger et al., 2014*). HIF increases KDM5C levels and activity, and the overall level of H3K4me3 is elevated when KDM5C is suppressed in *VHL*-defective kidney cancer cells (*Niu et al., 2012*). In addition, evidence suggests that HIF-induced KDM5C is tumor-suppressive, and this constitutes a negative feedback loop in regard to tumor growth (*Niu et al., 2012*).

SETD2 is a histone-modifying enzyme that tri-methylates histone H3 at Lysine 36 (H3K36me3). It is mutated in 10 – 15% of ccRCC tumors (*Dalgliesh et al., 2010*; *Cancer Genome Atlas Research Network, 2013*; *Sato et al., 2013*; *Hakimi et al., 2013*). It is also located on chromosome 3 p like *VHL* and *PBRM1*. *SETD2* mutations are subclonal in ccRCC tumors (*Gerlinger et al., 2012*; *Gerlinger et al., 2014*; *Sankin et al., 2014*), and are associated with worse patient survival (*Hakimi et al., 2013*). SETD2 deficiency was reported to be associated with alternative splicing and transcriptional repression (*Wagner and Carpenter, 2012*). Indeed, *SETD2* mutations in ccRCC tumors are associated with changes in chromatin accessibility and DNA methylation (*Buck et al., 2014*) or widespread RNA processing defects (*Simon et al., 2014*). Recently, SETD2 was shown to regulate interferon signaling by methylating STAT1 (*Chen et al., 2017*), and to maintain mitosis and cytokinesis through methylating α-tubulin (*Park et al., 2016*). The relevance of these SETD2 functions to tumor suppression in ccRCC remains to be elucidated.

Like KDM5C and SETD2, BAP1 is also a histone modifying enzyme. *BAP1* is inactivated in 10 – 15% of ccRCC (*Guo et al., 2012*; *Peña-Llopis et al., 2012*). The *BAP1* gene is located on chromosome 3 at band 3p21, which is very close to the *PBRM1* gene. As with *VHL*, *PBRM1* and *SETD2*, biallelic inactivation of *BAP1* via mutation and loss of heterozygosity fit a two-hit tumor suppressor

profile. Germline mutations of BAP1 predispose patients to ccRCC, uveal melanomas, mesothelioma, lung adenocarcinoma, meningioma, breast carcinoma and paraganglioma (*Abdel-Rahman et al., 2011*; *Wadt et al., 2012*; *Cheung et al., 2013*; *Pilarski et al., 2014*). Loss of BAP1 is associated with worse patient survival in ccRCC and seems to be mutually exclusive with *PBRM1* mutations (*Joseph et al., 2014*; *Joseph et al., 2016*). BAP1 protein was reported to form a complex with host cell factor-1 (HCF-1) and promotes DNA double-strand break repair (*Yu et al., 2010*; *Yu et al., 2014*). It also forms a complex with ASXL1 which is essential for its ability to deubiquitinate histone H2A on lysine 119, and this function regulates cell proliferation (*Daou et al., 2015*). Recently it was shown to regulate IP3R3-mediated $Ca^{2+}$ to mitochondria to suppress cell transformation (*Bononi et al., 2017*). How these functions contribute to its tumor suppressor role in ccRCC is unclear.

These secondary tumor suppressors each have their own distinct biological functions and are associated with different prognoses for the patients, and yet they all regulate chromatin biology. Thus, we investigated whether they share a common tumor suppressor pathway in this study.

## Results

### Interferon (IFN)-responsive genes are upregulated by PBRM1

To determine how the suppression of PBRM1 changes the transcriptome of ccRCC cells, we used shRNA to stably knock down the expression of PBRM1 in *VHL+/+* and *VHL-/-* 786-O cells. Then, we used Illumina HumanHT-12_V4 microarrays to compare gene expression between cells expressing a control shRNA (SCR) and PBRM1 shRNA-94. PBRM1 loss in *VHL-/-* cells elicited a higher number of transcriptional changes than in *VHL+/+* cells (*Figure 1A*). In addition, in *VHL-/-* cells, the transcriptional responses to PBRM1 depletion or VHL re-expression shared a group of gene targets (*Figure 1B*). This is consistent with the previous observation that PBRM1 loss amplifies a HIF signature (*Gao et al., 2017*; *Nargund et al., 2017*). Gene Ontology (GO) pathway analysis of the shared genes revealed that type I IFN signaling pathway and response to virus are commonly affected (*Figure 1C*). Gene Sets Enrichment Analysis (GSEA) confirmed that the expression signature of interferon alpha response was negatively correlated with PBRM1 suppression (*Figure 1D*). Gene expression analysis revealed the reduced expression of IFN-responsive genes after PBRM1 loss (*Figure 1E*).

### IFN-responsive genes are also maintained by KDM5C

KDM5C is a histone-modifying enzyme that is frequently mutated in ccRCC. In order to learn how the loss of KDM5C impacts the transcriptome of ccRCC cells, we stably knocked down the expression of KDM5C in *VHL+/+* and *VHL-/-* 786-O cells. Then we compared gene expression between cells expressing a control shRNA (SCR) and KDM5C shRNA-60 with the methods described above. Similar to PBRM1 suppression, KDM5C loss in *VHL-/-* cells elicited greater transcriptional changes than that in *VHL+/+* cells (*Figure 2A*). We also observed that in *VHL-/-* cells, transcriptional responses to KDM5C depletion or VHL re-expression shared a group of gene targets (*Figure 2B*). This is consistent with the previous observation that HIF activates KDM5C and this constitutes a negative feedback loop (*Niu et al., 2012*). GO pathway analysis, GSEA, and gene expression data confirmed that the type I IFN signaling pathway, the expression signature of interferon alpha response, and the IFN-responsive genes were impacted by KDM5C loss (*Figure 2C–E*).

### Suppression of KDM5C and PBRM1 have similar impacts on gene expression

Since the suppression of either PBRM1 or KDM5C in *VHL-/-* 786-O cells significantly impacted the type I IFN signaling pathway (*Figures 1B* and *2B*), and the gene lists largely overlapped (*Figures 1E* and *2E*), we compared the transcriptional responses to the loss of PBRM1 or KDM5C. The vast majority of the upregulated genes and downregulated genes after suppression of PBRM1 or KDM5C were shared (*Figure 3A*). The GO pathway analysis of the shared genes that are suppressed by PBRM1 or KDM5C did not reveal any significantly impacted pathway, but the shared induced genes showed that "response to virus" was significantly enriched (*Figure 3B*). The heat map shows the commonly affected genes (*Figure 3C*). To confirm that the transcriptional response was not the

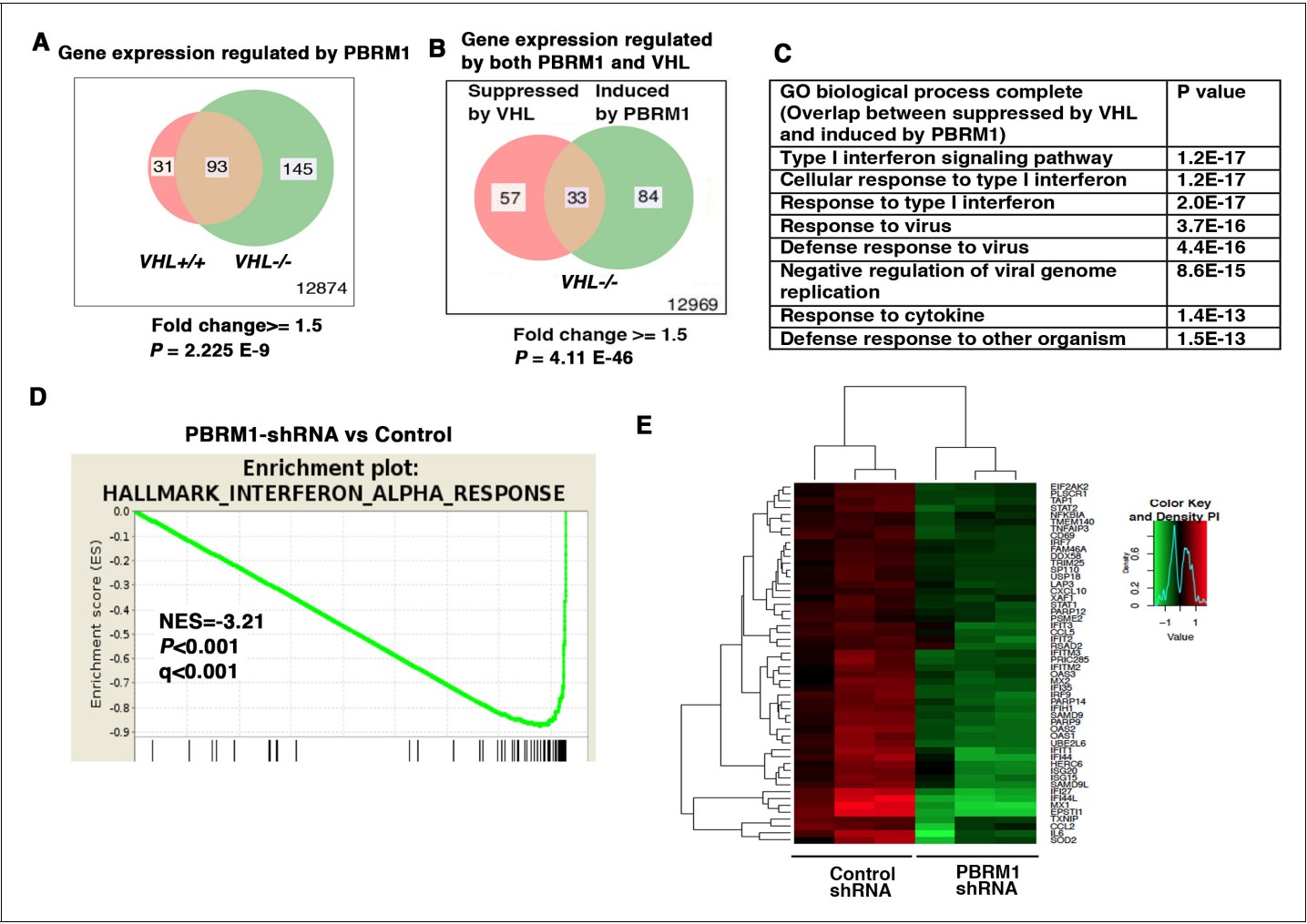

**Figure 1.** IFN-responsive genes were downregulated by VHL and upregulated by PBRM1. (A) Venn diagram showing that PBRM1 loss had greater impact on transcriptome in *VHL-/-* cells than that in *VHL+/+* cells; (B) Venn diagram showing the overlap of genes downregulated by VHL and upregulated by PBRM1 in 786-O cells. *P*-values were calculated with hypogeometric probability. (C) Enriched gene ontology (GO) analysis of genes regulated both by VHL and PBRM1. (D) GSEA analysis of genes upregulated by PBRM1. (E) Heatmap showing the expression of IFN-responsive genes after PBRM1 knockdown.

result of the off-target effects of a single shRNA, we suppressed the expression of PBRM1 or KDM5C with two shRNA constructs each. Then we measured the expression of selected IFN-responsive genes with RT-PCR, and found that the transcriptional changes in *IL6*, *ISG20*, *IL8*, *ARPC4*, *GAS6*, and *STMN3* were consistent (*Figure 3D*). To ensure that our observation is not confined to just one cell line, we performed similar experiments in Ren-02 cells, another VHL-/- ccRCC cell line. Again, the suppression of PBRM1 or KDM5C with different shRNA constructs produced consistent transcriptional changes (*Figure 3E*), suggesting that PBRM1 loss or KDM5C loss elicits similar transcriptional responses.

## PBRM1 interacts with KDM5C

Since loss of PBRM1 or KDM5C elicited very similar transcriptional responses, we examined whether these two proteins exist in one complex. When both proteins are expressed in HEK293T cells, immunoprecipitation of either protein pulled down the other (*Figure 4A*). In addition, over-expressed Flag-PBRM1 was able to immunoprecipitate endogenous KDM5C protein and other PBAF subunits (*Figure 4B*). Flag-KDM5C was able to pull down endogenous PBRM1 along with BRG1 and BAF170, the catalytic and structural subunits of PBAF complex respectively (*Figure 4C*). Interestingly, it did not significantly co-IP BRD7 or BAF57, unlike that of Flag-PBRM1 (*Figure 4B*), suggesting that Flag-

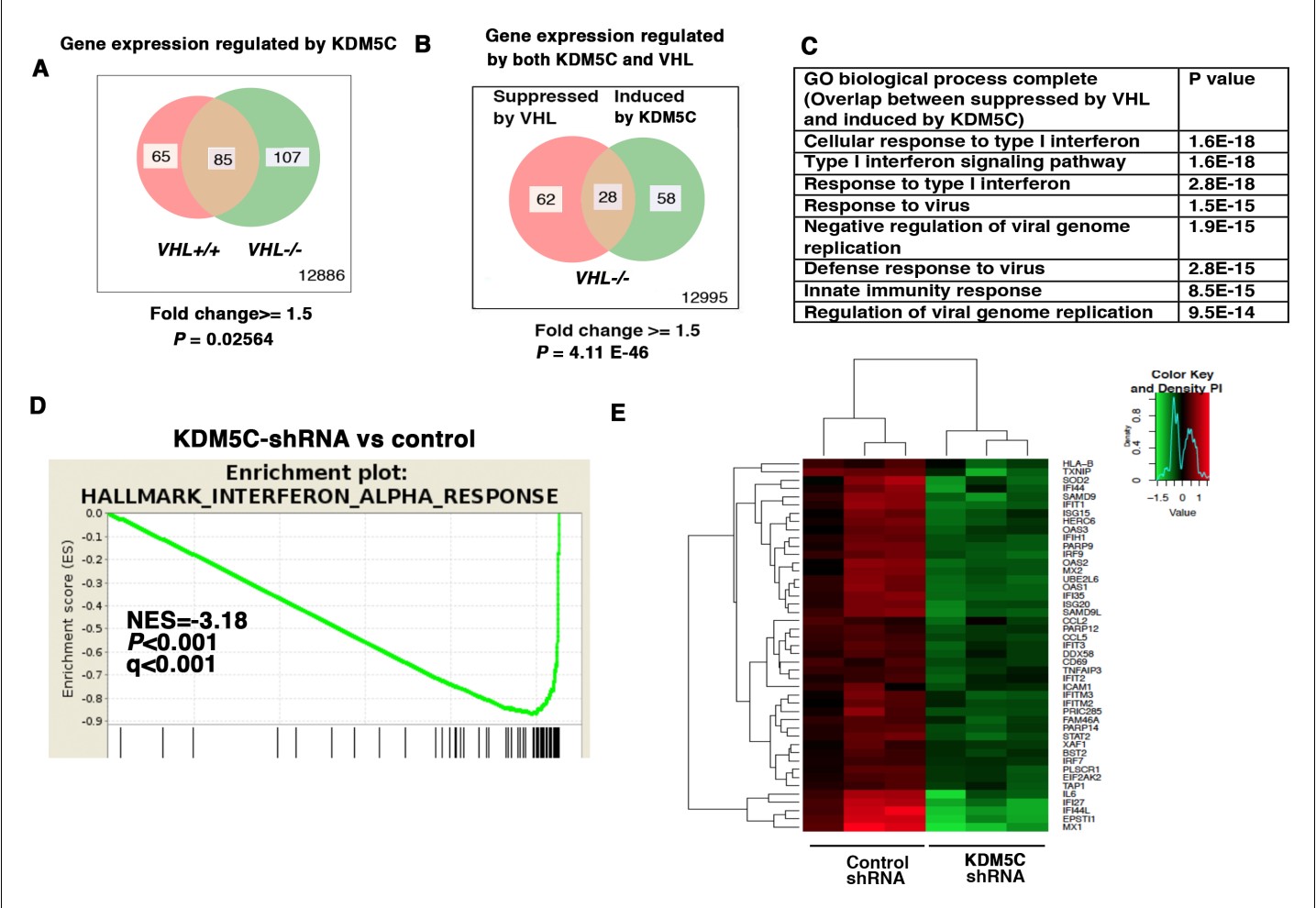

**Figure 2.** IFN-responsive genes were downregulated by VHL and upregulated by KDM5C. (**A**) Venn diagram showing that KDM5C loss had greater impact on transcriptome in *VHL-/-* cells than that in *VHL+/+* cells; (**B**) Venn diagram showing the overlap of genes downregulated by VHL and upregulated by KDM5C in 786-O cells. *P*-values were calculated with hypogeometric probability. (**C**) Enriched gene ontology (GO) analysis of genes regulated both by VHL and KDM5C. (**D**) GSEA analysis of genes upregulated by KDM5C. (**E**) Heatmap showing the expression of IFN-responsive genes after KDM5C knockdown.

KDM5C pulled down a partial PBAF complex. Moreover, the endogenous PBRM1 also pulled down the endogenous KDM5C in human embryonic kidney (HEK) 293T cells along with the other PBAF subunits (*Figure 4D*), suggesting that their interaction is not an artifact due to over-expression. PBRM1 truncation analysis suggested that its C-terminus, possibly the HMG domain, is important for KDM5C interaction (*Figure 4E*). These results suggest that two proteins exist in one protein complex, and this could be the mechanism that they regulate the expression of similar target genes.

## VHL, PBRM1 and KDM5C share the regulation of ISGF3

Since loss of both PBRM1 and KDM5C triggered greater transcriptional responses in *VHL-/-* cells than that in *VHL+/+* cells, and HIF transcriptionally interacts with PBRM1 and KDM5C (*Gao et al., 2017*; *Niu et al., 2012*), we wondered whether VHL-regulated genes overlapped with PBRM1 and KDM5C-regulated genes. Comparison of the transcriptional changes revealed that a group of 27 genes was suppressed by VHL but induced by PBRM1 or KDM5C (*Figure 5A*), and these genes were the interferon-responsive genes (*Figure 5B*). A GO pathway analysis showed that these shared genes belong to the type I interferon signaling pathway (*Figure 5C*).

Close inspection of these genes suggested that they are the transcriptional targets of interferon stimulated gene factor 3 (ISGF3). ISGF3 is a heterotrimeric transcription factor composed of STAT1,

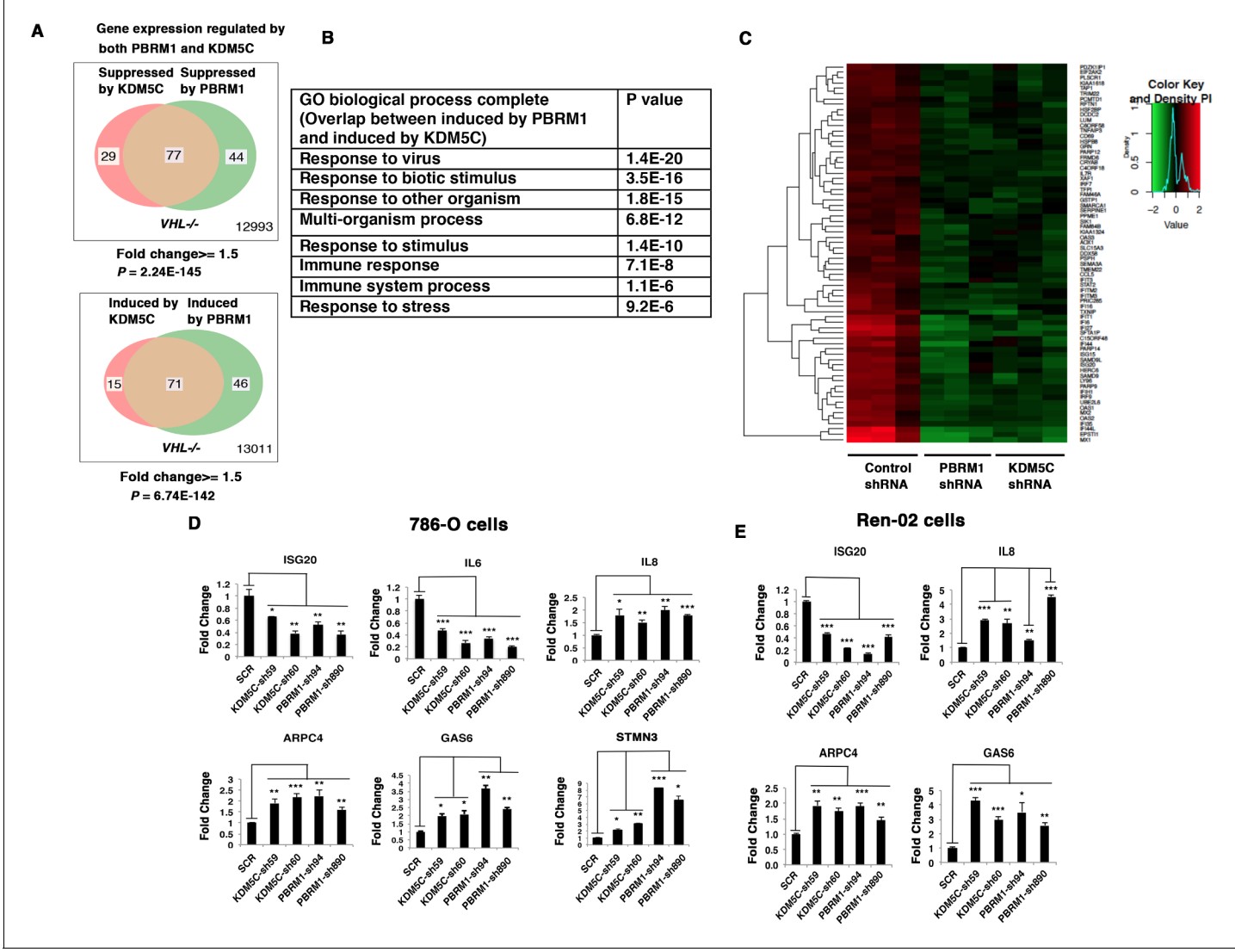

**Figure 3.** Suppression of KDM5C or PBRM1 had similar impact on gene expression in ccRCC cells. (**A**) Venn diagrams showing the overlaps of genes whose expression are affected by PBRM1 or KDM5C losses in 786-O. The p-value significance of the overlap in a Venn diagram was calculated using hypergeometric test in R; (**B**) Enriched gene ontology (GO) analysis of genes regulated both by PBRM1 and KDM5C; (**C**) Heat map of shared gene targets by PBRM1 and KDM5C; (**D-E**) Examination of mRNA expression of individual genes by RT-qPCR in 786-O (**D**) and Ren-02 (**E**) after PBRM1 or KDM5C was suppressed by shRNA. The p-value significance in the bar graph was calculated with two-tailed student *t* test. *p<0.05; **p<0.01; ***p<0.001. analysis of the shared genes that are suppressed by PBRM1 or KDM5C did not reveal any significantly impacted pathway, but the shared induced genes showed that 'response to virus' was significantly enriched (***Figure 3B***). The heat map shows the commonly affected genes (***Figure 3C***). To confirm that the transcriptional response was not the result of the off-target effects of a single shRNA, we suppressed the expression of PBRM1 or KDM5C with two shRNA constructs each. Then, we measured the expression of selected IFN-responsive genes with RT-PCR, and found that the transcriptional changes in *IL6, ISG20, IL8, ARPC4, GAS6,* and *STMN3* were consistent (***Figure 3D***). To ensure that our observation is not confined to just one cell line, we performed similar experiments in Ren-02 cells, another *VHL-/-* ccRCC cell line. Again, the suppression of PBRM1 or KDM5C with different shRNA constructs produced consistent transcriptional changes (***Figure 3E***), suggesting that PBRM1 loss or KDM5C loss elicits similar transcriptional responses.

STAT2, and IRF9, and it can be activated either by interferon-induced posttranslational modifications or increased protein levels (***Cheon et al., 2013***). Since these genes are suppressed by VHL, components of ISGF3 might be HIF targets. To test this, we suppressed HIF2α expression with two shRNA constructs in 786-O cells, then examined the expression of the ISGF3 components. After HIF2α suppression, the protein levels of IRF9 and STAT2 were reduced while the STAT1 protein level did not change (***Figure 5D***). Conversely, transient expression of a stable and active form of HIF2α in *VHL+/*

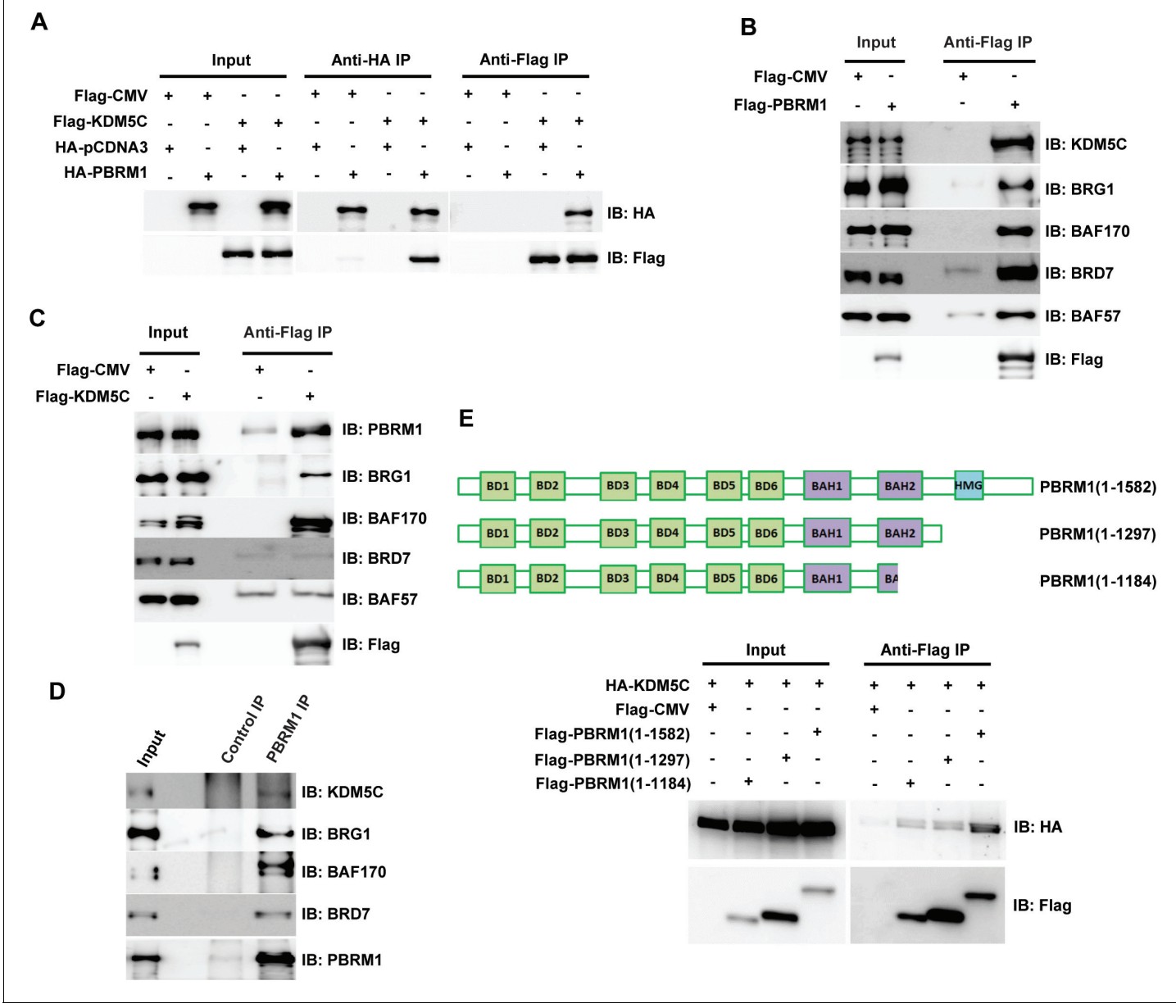

**Figure 4.** PBRM1 interacts with KDM5C. (**A**) The indicated plasmids were transfected into HEK293T cells. Immunoprecipitations were performed followed by immunoblots with the indicated antibodies. Flag-PBRM1 (**B**) or Flag-KDM5C (**C**) was transfected into HEK293T cells. Immunoprecipitations were performed followed by immunoblots with the indicated antibodies. (**D**) HEK293T cell lysate was immunoprecipitated with indicated antibodies followed by immunoblots with the indicated antibodies. (**E**) A schematic diagram of PBRM1 constructs used for KDM5C interaction analysis (top). The indicated plasmids were transfected into HEK293T cells. Immunoprecipitations were performed followed by immunoblots with the indicated antibodies (bottom).

+ccRCC cell lines, Caki-1 and Ren-01, slightly increased the expression levels of ISGF3 and its targets (*Figure 5—figure supplement 1*). Suppression of HIF1α did not reduce ISGF3 protein level or function in RCC4 cells (*Figure 5—figure supplement 2*). To test whether PBRM1 and KDM5C also had similar effects on ISGF3, we examined its components in 786-O cells with PBRM1 and KDM5C knocked down with various shRNA constructs. In PBRM1 knockdown cells, the protein levels of IRF9 and STAT2 were lower compared to control (*Figure 5D*). In KDM5C knockdown cells, the protein level of STAT2 was lower (*Figure 5D*). Consistent with this, the expression of ISGF3 target genes were significantly reduced in 786-O cells after PBRM1 or KDM5C suppression (*Figure 5E*).

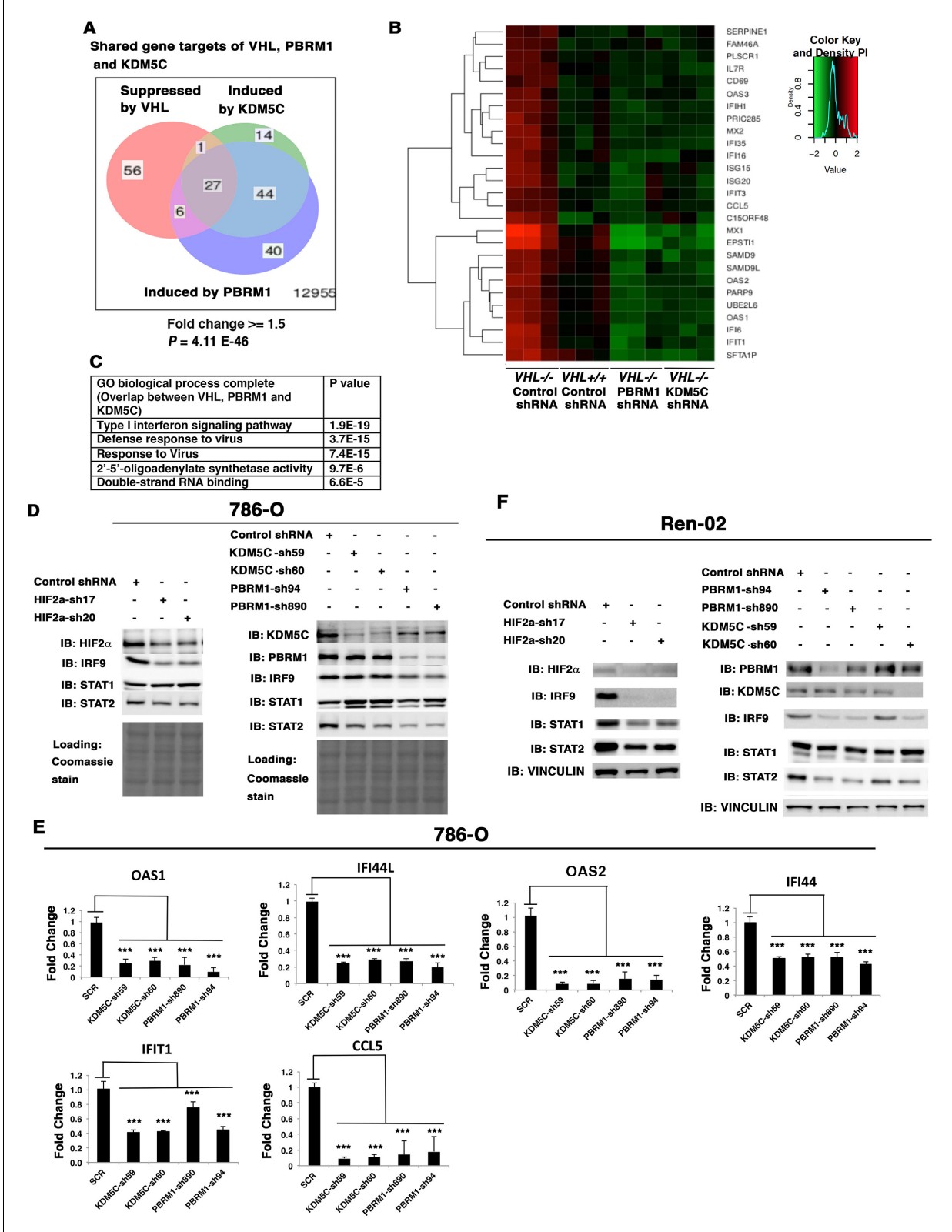

**Figure 5.** HIF, PBRM1 and KDM5C are required for the expression and activity of ISGF3. A Venn diagram showing the overlaps of genes whose expression was affected by VHL, PBRM1 or KDM5C losses in 786-O cells. The p-value significance of the overlap in a venn diagram was calculated using hypergeometric test in R; (B) Heat map of shared gene targets by VHL, PBRM1 and KDM5C; (C) Enriched gene ontology (GO) analysis of genes regulated both by VHL, PBRM1 and KDM5C; (D) Western blots with the indicated antibodies in 786-O cell SDS-solubilized whole cell lysates with HIF2α,

*Figure 5 continued on next page*

*Figure 5 continued*

PBRM1 or KDM5C suppressed by shRNA; (E) Examination of mRNA expression of individual genes by RT-qPCR in 786-O cells after PBRM1 or KDM5C suppression by shRNA; (F) Western blots with the indicated antibodies in Ren-02 cell SDS-solubilized whole cell lysates with HIF2α, PBRM1 or KDM5C suppressed by shRNA. The p-value significance in the bar graph was calculated with two-tailed student t test. *p<0.05; **p<0.01; ***p<0.001.

The online version of this article includes the following figure supplement(s) for figure 5:

**Figure supplement 1.** Over-expression of HIF2α in VHL+/+ccRCC cells increases ISGF3 levels and activity.

**Figure supplement 2.** Suppression of HIF1α does not reduce ISGF3 protein level or function in RCC4 cells.

**Figure supplement 3.** The reduced expression of ISGF3 targets in Ren-02 cells after suppression of PBRM1 or KDM5C.

**Figure supplement 4.** The ISGF3 is mostly unphosphorylated in 786-O ccRCC cells.

To ensure that this observation is not confined to just one ccRCC cell line, we knocked down HIF2α, PBRM1 or KDM5C in another VHL-deficient ccRCC cell line, Ren-02. After HIF2α suppression, all three subunits of ISGF3 showed reduced expression (*Figure 5F*). After PBRM1 or KDM5C suppression, the protein levels of IRF9 and STAT2 dropped (*Figure 5F*). Consistent with this, the expression of ISGF3 target genes was significantly reduced in Ren-02 cells after PBRM1 or KDM5C suppression (*Figure 5—figure supplement 3*).

ISGF3 can be activated by either interferon-induced phosphorylation on STAT1 and STAT2 or increased unphosphorylated protein levels (*Cheon et al., 2013*; *Borden et al., 2007*). To determine the phosphorylation status of ISGF3, we used interferon α to stimulate 786-O cells with or without PBRM1 (*Figure 5—figure supplement 4A*) or KDM5C (*Figure 5—figure supplement 4B*) knocked down and found that the ISGF3 was mostly unphosphorylated. This suggests that the elevated protein levels of ISGF3 components are responsible for the enhanced activity of ISGF3.

## SETD2 and BAP1 also regulate the expression and function of ISGF3

A recent report suggested that SETD2 promotes ISGF3 activity through methylating STAT1 after interferon stimulation (*Chen et al., 2017*). This prompted us to examine whether the remaining major tumor suppressors in ccRCC, SETD2 and BAP1, also regulate ISGF3 in *VHL-/-*ccRCC cells. We knocked down the expression of SETD2 and BAP1 with shRNA constructs in Ren-02 cells and observed significant down-regulation of IRF9 and STAT2 (*Figure 6A*, and *Figure 6—figure supplement 1*). In addition, we observed reduced expression of PLSCR1 and MX1, two ISGF3 targets. BAP1 re-expression in a Ren-02 BAP1 knockdown (*Figure 6B*, left) or knockout clone (*Figure 6B*, right) generated by CRISPR/Cas9 increased the protein levels of STAT2, IRF9 and PLSCR1, suggesting that BAP1's impact on ISGF3 is specific. To ensure that this effect is not confined to one cell line, we knocked down the expression of BAP1 or SETD2 in RCC4 and BAP1 in A498 cells (A498 has an inactivating mutation in *SETD2* so it is not suitable for SETD2 analysis), two additional VHL-deficient ccRCC cell lines. Their suppression also led to the reduced expression of IRF9, STAT2, PLSCR1 and MX1 (*Figure 6C and D*). Again, the impact of BAP1 loss in A498 cells was reversed by BAP1 re-expression (*Figure 6E*). In order to further examine the impact of BAP1 on ISGF3 expression and activity, we expressed GFP, wild type BAP1, or BAP1 with tumor-derived mutations (N78S or G185R) in BAP1 null ccRCC cell lines UMRC2 and UMRC6. Expression of wild type or N78S BAP1 clearly increased ISGF3 protein levels and its activity in these cells, while the G185R mutant failed to do so (*Figure 6F and G*). Thus SETD2 and BAP1 also are required to maintain the expression and function of ISGF3 in multiple VHL-deficient ccRCC cell lines. As a ccRCC tumor-derived mutation in BAP1 disrupted this function in both BAP1-null cells, it suggests that the regulation of ISGF3 is important to BAP1's tumor suppressor function.

## ISGF3 is tumor suppressive in a xenograft model

ISGF3 is known to be involved in the DNA damage response, response to immunotherapies and, canonically, the response to viral infection (*Borden et al., 2007*). There is no report showing its direct involvement in regulating tumorigenesis or tumor growth. Since ISGF3 is activated by HIF and suppressed after the loss of PBRM1, KDM5C, SETD2 or BAP1, it may function as the executioner of a negative feedback loop that potently opposes tumor growth. To test this, we first confirmed that IRF9-sh69 shRNA construct successfully downregulated IRF9 expression (*Figure 7A*). Suppression of IRF9 did not significantly change cell growth in culture (*Figure 7B*). We injected the same number of

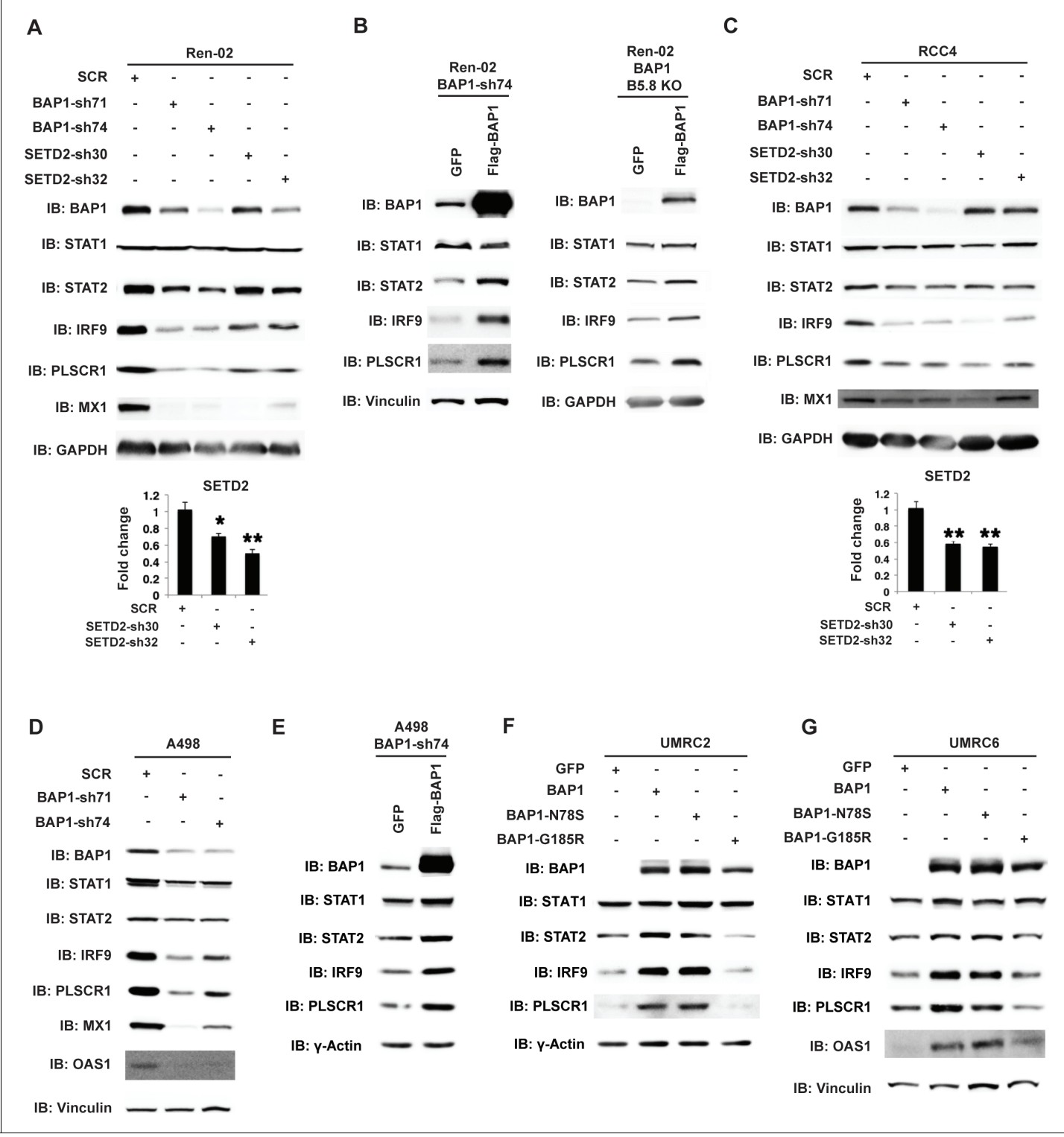

**Figure 6.** BAP1 and SETD2 are required for the expression and activity of ISGF3. SDS-solubilized whole cell lysates of Ren-02 cells expressing control shRNA or shRNA against BAP1 or SETD2 were blotted with the indicated antibodies.(**B**) Control vector expressing GFP or BAP1 were stably transfected into Ren-02 cells expressing BAP1-sh74 (left). The BAP1 plasmid carries silent mutations against BAP1-sh74 to cancel the effect of shRNA. Control vector expressing GFP or BAP1 were transiently transfected into a Ren-02 clone with BAP1 knocked out by CRISPR/Cas9 (right). The SDS-solubilized whole cell lysates were blotted with the indicated antibodies. C-D) SDS-solubilized whole cell lysates of RCC4 cells (**C**) or A498 cells (**D**) expressing the indicated shRNA constructs were blotted with the indicated antibodies. The knockdown efficiency of SETD2 was measured with RT-PCR in panels A and C. (**E**). Control vector expressing GFP or BAP1 with silent mutations were stably transfected into A498 cells expressing BAP1-sh74. BAP1-null UMRC2 (**F**)

*Figure 6 continued on next page*

*Figure 6 continued*

or UMRC6 (**G**) ccRCC cells were stably transfected with indicated plasmids. The SDS-solubilized whole cell lysates were blotted with the indicated antibodies.

The online version of this article includes the following figure supplement(s) for figure 6:

**Figure supplement 1.** Suppression of SETD2 expression by western blot.

786-O cells expressing control shRNA (SCR) or IRF-sh69 into the flanks of nude mice. The 786-O cells with IRF9 suppression produced much larger tumors than the control cells (*Figure 7C and D*). Knockdown of IRF9 in Ren-02 cells also significantly boosted tumor growth in a xenograft model (*Figure 7—figure supplement 1*), IRF9 suppression did not reduce the expression of other subunits of ISGF3 (*Figure 7—figure supplement 2A*), and IRF9 suppression was maintained in the endpoint tumors (*Figure 7—figure supplement 2B*). Furthermore, we stably suppressed the expression of either STAT1 or STAT2 in 786-O cells then performed xenograft analysis. In both cases, the tumors were significantly bigger than the ones made by the control cells (*Figure 7—figure supplement 3*). These data strongly suggest that ISGF3 has potent anti-tumor activity *in vivo*.

In order to learn the reason for enhanced tumor growth after IRF9 knockdown, we performed hematoxylin and eosin(H&E) and immunohistochemistry (IHC) staining on five pairs of 786-O SCR and IRF9-69 tumors. The control tumors contained higher percentages of mouse stromal cells, while the IRF9 knockdown tumors were mostly comprised of cancer cells (*Figure 7E* and *Figure 7—figure supplement 4*). The cancer cells in tumors with IRF9 knockdown had a slight but insignificant increase in Ki67 staining, a marker for cell proliferation, over cancer cells in control tumors (*Figure 7E*). However, CD45 staining, a marker for all mouse immune cells, was significantly higher in control tumors compared to IRF9 knockdown tumors (*Figure 7E*, p=0.007). There were no significant differences in cleaved caspase three or CD31, markers for apoptosis or blood vessels, respectively (*Figure 7—figure supplement 4*). Thus the lack of infiltrating host immune cells into the tumors formed by ccRCC cells depleted of IRF9, not changes in cell proliferation, apoptosis, or blood vessel growth, might be the primary reason of enhanced growth.

## Over-expression of ISGF3 in PBRM1-deficient ccRCC cells strongly suppresses tumor growth

Suppression of IRF9 in 786-O or Ren-02 cells greatly enhanced tumor growth (*Figure 7* and *Figure 7—figure supplement 1*). As PBRM1 suppression led to reduced ISGF3 expression and activity (*Figure 5*) and enhanced tumor growth (data not shown), we asked whether ISGF3 over-expression reverses the enhanced tumor growth elicited by PBRM1 deficiency. To this end, we stably infected 786-O PBRM1-sh94 cells with different combinations of viral vectors expressing STAT1, STAT2 or IRF9. The combination of STAT2 +IRF9 or all three factors strongly induced ISGF3 targets PLSCR1 and MX1 (*Figure 8A*) and we chose the cells expressing STAT2 +IRF9 for further analysis. The overexpression of STAT2 +IRF9 did not change cell growth in culture (*Figure 8B*), however, overexpression of these ISGF3 components strongly suppressed tumor growth by PBRM1-deficient 786-O cells in a xenograft model (*Figure 8C and D*).Thus depletion of ISGF3 greatly enhanced tumor growth while its activation very potently blocked it, suggesting that ISGF3 is a central player in ccRCC tumor growth that is targeted by most of the cancer genes.

## IRF9 and STAT2 expression significantly correlated with the expression of PBRM1, SETD2, or BAP1 in human ccRCC tumors

To verify that the observations are relevant for human ccRCC, we analyzed TCGA ccRCC datasets to investigate possible connection between ISGF3 and the secondary tumor suppressors. However, we did not find consistent correlations between mRNA levels of BAP1, PBRM1, KDM5C, SETD2 and ISGF3 components in the whole tumors (data not shown). One possibility for these results is that the mRNA expression of ISGF3 components detected in TCGA is not primarily coming from cancer cells, but from other stromal cells which can distort the results. To investigate this, we analyzed the correlations between well-known immune and tumor cell markers (CD45 or PTPRC, CD3, CD8, etc and CDH1, EPCAM) with ISGF3 subunits within the TCGA dataset. We found that the mRNA expression of ISGF3 subunits in the TCGA dataset comes predominantly from immune cells as opposed to

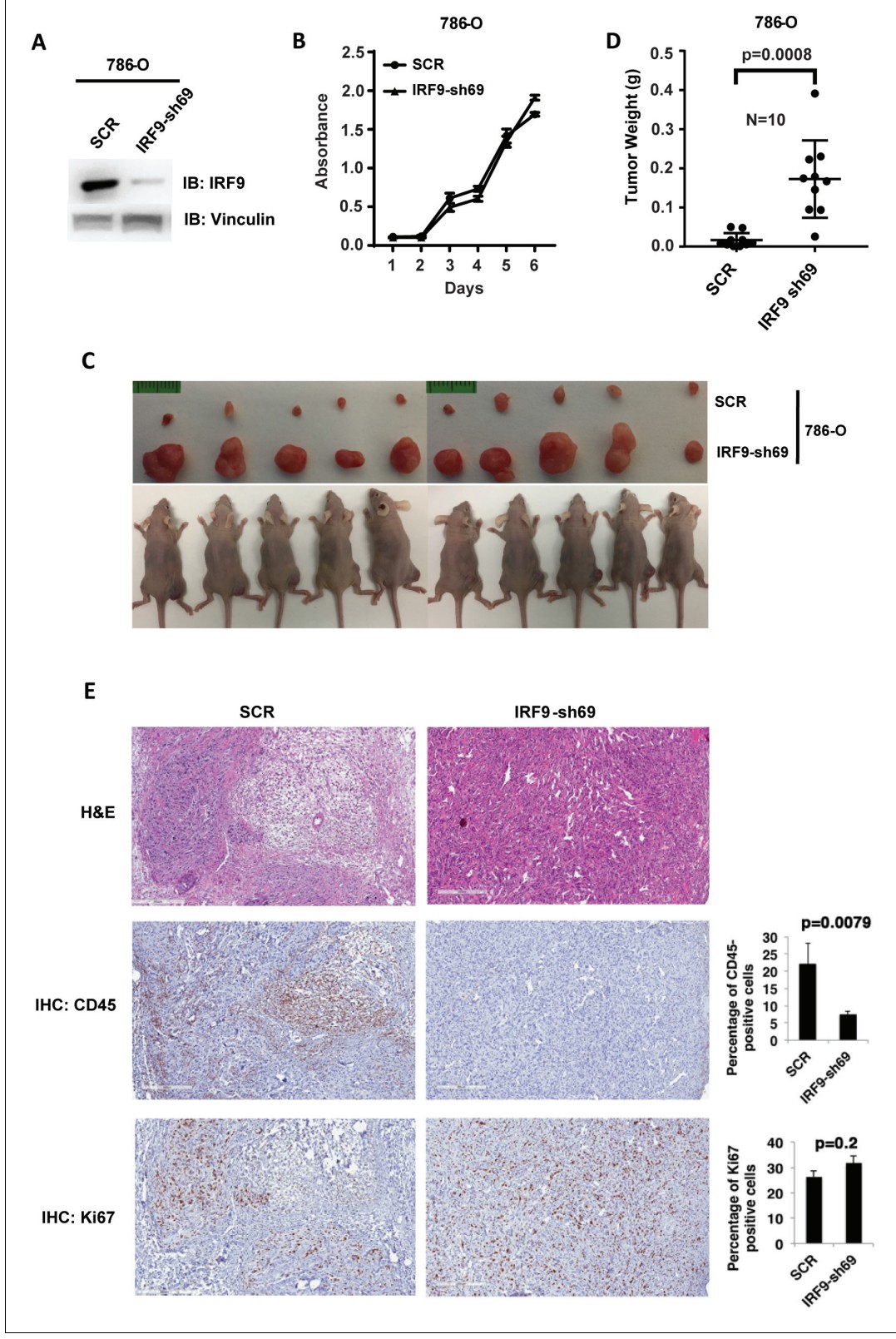

**Figure 7.** Suppression of IRF9, a key component of ISGF3, led to enhanced tumor growth and significantly less tumor-infiltrating immune cells. (A) SDS-solubilized whole cell lysates of 786-O cells expressing control shRNA or shRNA against IRF9 were blotted with indicated antibodies; (B) The growth rate of the indicated cell lines measured by WST-1 assay; (C) Pictures of mice and xenografted tumors and (D) Quantification of tumor weights
*Figure 7 continued on next page*

*Figure 7 continued*

originating from same number of cancer cells described in (**A**) in athymic nude mice. The p-value was calculated with two-tailed student t test; (**E**) H&E or immunohistochemistry stains of slides from the representative pair five tumors. The percentages of CD45- or Ki67-positive cancer cells in the tumors were quantified with analytic programs in Aperio software. The p-values were calculated with Mann Whitney-Wilcoxon.

The online version of this article includes the following figure supplement(s) for figure 7:

**Figure supplement 1.** Suppression of IRF9 significantly enhanced tumor growth by Ren-02 cells.

**Figure supplement 2.** Stable suppression of IRF9 in ccRCC cells and xenograft tumors.

**Figure supplement 3.** Suppression of STAT1 or STAT2 promotes tumor growth.

**Figure supplement 4.** IRF9 loss does not lead to significant changes in apoptosis or blood vessel density.

tumor cells (surrogated by EPCAM and E cadherin) (*Figure 9—figure supplement 1*). We also sought to confirm this with immunohistochemistry (IHC). We found that immune cells had much higher protein expression of ISGF3 subunits STAT2 than the normal or cancerous human kidney cells. On the same slide, the immune cells in lymph node, spleen and lung tissues were stained strongly positive with antibody against STAT2, but the normal and cancerous kidney tissue stained negative or very weakly (*Figure 9—figure supplement 2A*). Interestingly, in the lung tissue, the strong staining of STAT2 in the immune cells did not seem to significantly increase the staining intensity of the surrounding cells. Similar observation was made with IRF9 IHC (*Figure 9—figure supplement 2B*). Thus immune cells have much higher expression of ISGF3 subunits STAT2 and IRF9 than that in kidney cancer cells, and this could disrupt the detection of ISGF3 mRNA expression in the cancer cells. Since our hypothesis is that BAP1, PBRM1, KDM5C, SETD2 are critical to maintain the expression and/or function of ISGF3 subunits in the ccRCC cancer cells, we concluded that mRNA levels in the bulk tumors from the TCGA dataset are not suitable for our validation experiments.

To validate our hypothesis, we decided to examine the ISGF3 protein levels in ccRCC cancer cells with IHC and correlate their protein levels with the protein levels of BAP1, PBRM1, and SETD2. We have previously examined the protein levels of PBRM1 and SETD2 in a ccRCC tissue microarray (TMA) generated by our colleagues at Fox Chase Cancer Center in our previous publication (*Jiang et al., 2016*). The specificity of the BAP1 (*Figure 6A*), STAT2 (*Figure 7—figure supplement 3*) and IRF9 (*Figure 9—figure supplement 3A*) antibodies used for IHC was validated by western blots. We probed the same TMA with antibodies against BAP1, IRF9 and STAT2. The staining of BAP1 or IRF9 mostly occurs in the nucleus, while STAT2 staining is found in both cytoplasm and nucleus (*Figure 9—figure supplement 3B*). In many cases the protein expression levels of PBRM1, SETD2 or BAP1 correlated with that of STAT2 or IRF9 on the same tumor samples (*Figure 9A*). After scoring the samples and performing statistical analysis, the Spearman association analysis revealed that nuclear IRF9 and cytoplasmic STAT2 both significantly correlated with the expression of PBRM1, SETD2, or BAP1 (*Figure 9B*). The correlations were mostly statistically significant when all foci were considered and in individual tumor stage or grade. Interestingly, the loss of nuclear staining of IRF9 was associated with worse patient survival, consistent with a tumor suppressor role of ISGF3 (*Figure 9C*). The loss of cytoplasmic STAT2 staining was not associated with worse patient survival (*Figure 9D*), suggesting that other ISGF3 components are the more dominant players in the human ccRCC. Consistent with this idea, the change of IRF9 protein in the TMA (*Figure 9B*) and in the cultured cells (*Figures 5* and *6*) tracked better with the changes of PBRM1, SETD2 or BAP1. Taken together, our data suggest that the links between PBRM1, SETD2 or BAP1 and ISGF3 are preserved in human ccRCC tumors.

## Discussion

Although the overall activity of HIF is strongly oncogenic, our previous results indicate that it also triggers a significant anti-tumor response. For instance, KDM5C is induced by HIF, and its demethylase activity against global H4K4me3 mark is dependent on HIF in *VHL-/-*ccRCC cells, and yet it clearly plays an anti-tumor role in ccRCC (*Niu et al., 2012*). In addition, some other HIF downstream targets, such as VEGF or Cyclin D1, are strongly pro-tumor growth, whereas other HIF downstream targets such as GLUT1 or IGFBP3, are obviously tumor suppressive (*Zhang et al., 2013*). Consistent with this, *Vhl* conditional knockout in mouse kidneys alone is not sufficient to generate ccRCC

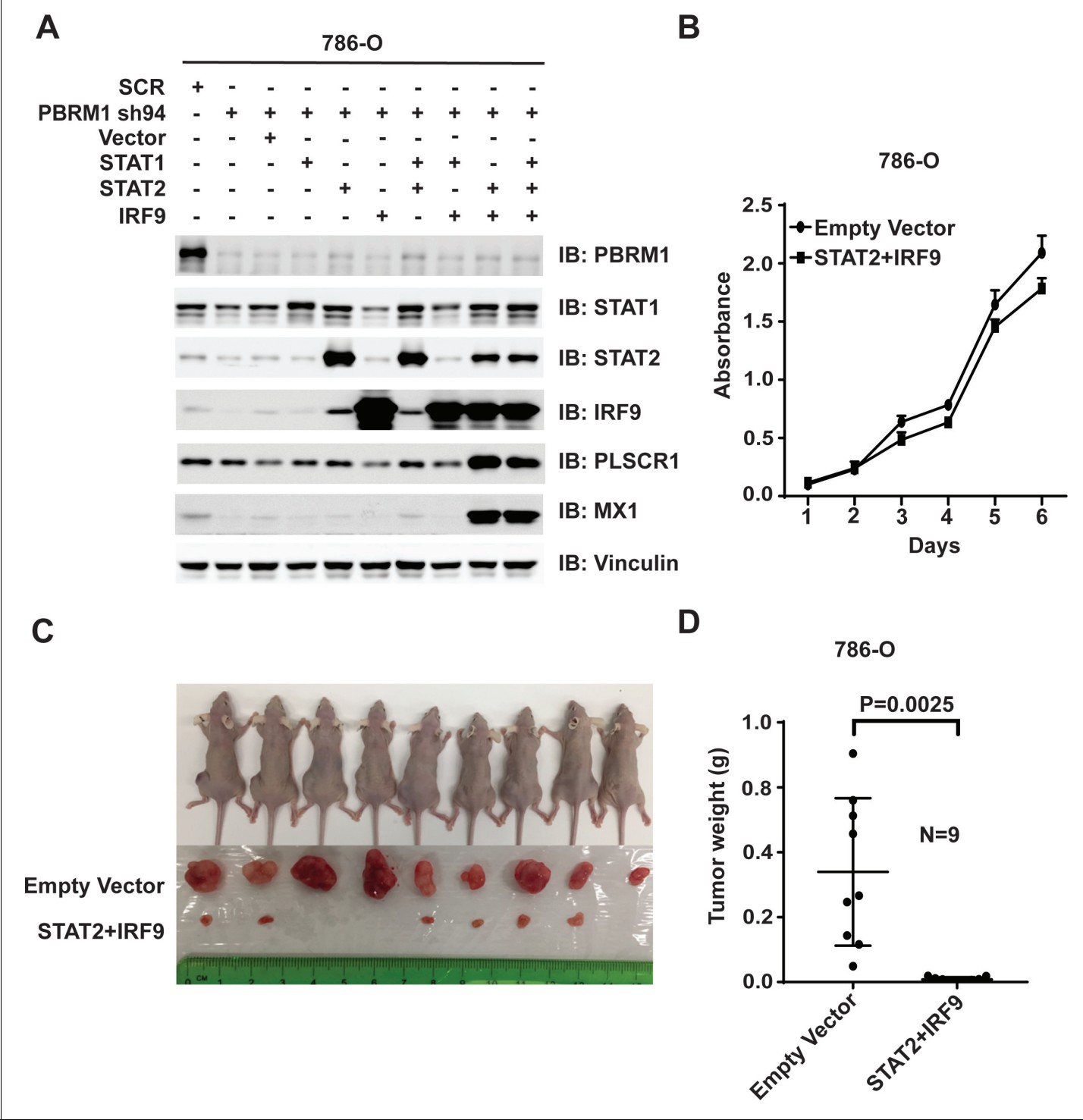

**Figure 8.** Over-expression of IRF9 and STAT2 restored ISGF3 function and retarded tumor growth of 786-O cells with PBRM1 suppressed. (A) 786-O cells were infected with the indicated viral vectors. SDS-solubilized whole cell lysates were blotted with the indicated antibodies. (B) Cellular growth of indicated cell lines measured by WST-1 assay; (C) Pictures of mice and xenografted tumors and (D) Quantification of tumor weights originating from the same number of cancer cells expressing either an empty vector or STAT2 plus IRF9 in athymic nude mice. The p-value was calculated with two-tailed student t test.

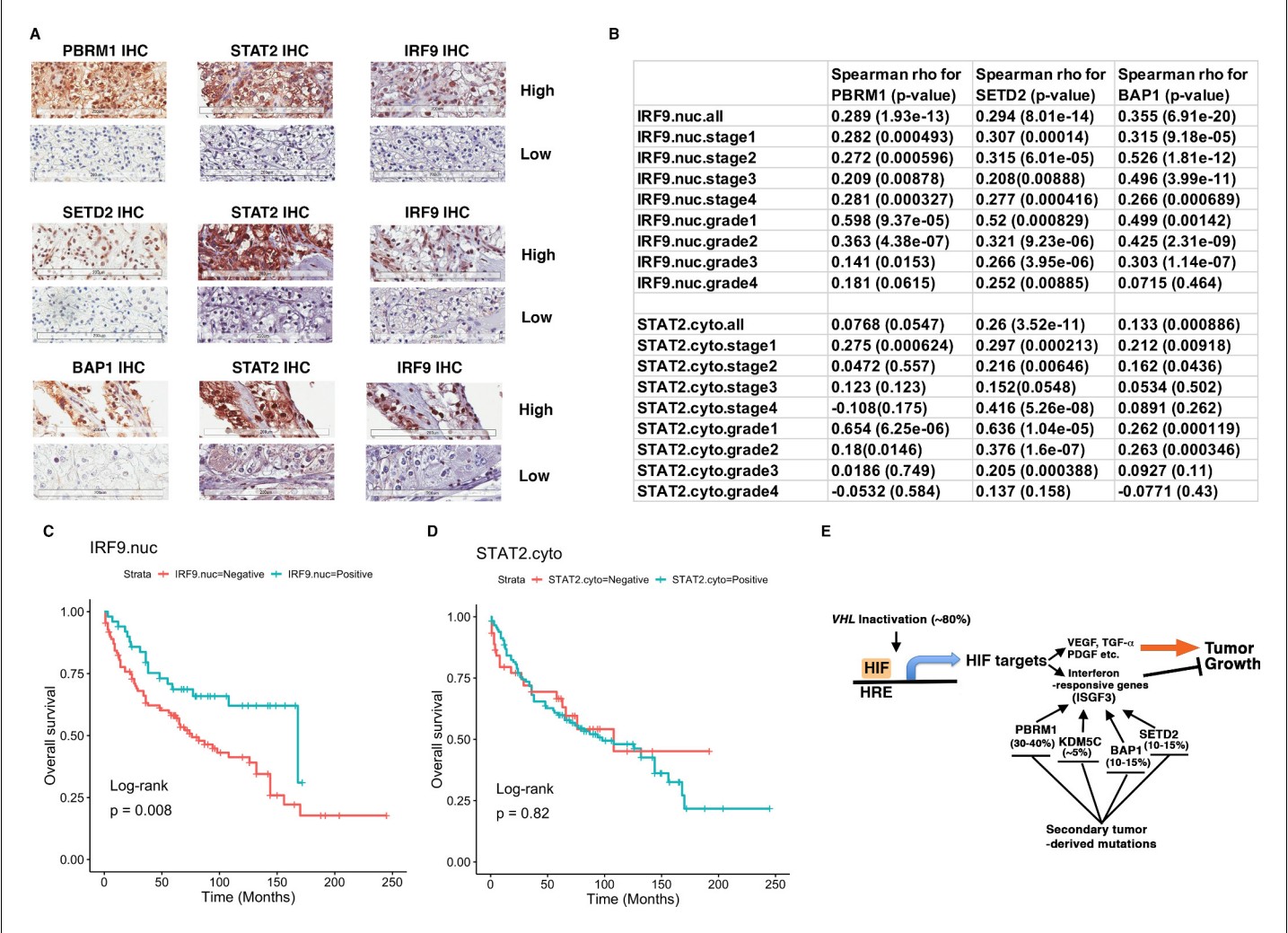

**Figure 9.** IRF9 and STAT2 expression significantly correlated with the expression of PBRM1, SETD2, or BAP1 in ccRCC tumors. (A) 20x IHC images of representative ccRCC foci stained with indicated antibodies. (B) Spearman correlation coefficient, rho, and p-value of nuclear (nuc) IRF9 or cytoplasmic (cyto) IHC signals with those of PBRM1, SETD2 or BAP1. The analysis was performed with either all the foci or the foci within each tumor stage or tumor grade. (C) Kaplan-Meier curves of patient overall survival based on the positive or negative IHC signal of nuclear IRF9 and (D) cytoplasmic STAT2. Statistical significance calculated using the log-rank test. (E) A schematic diagram showing how HIF and secondary tumor suppressors converge on ISGF3 to regulate tumor growth in ccRCC. HIF activates ISGF3 that suppresses tumor growth. PBRM1, KDM5C, SETD2 and BAP1 all support the function of ISGF3. Mutations in any of them relieve the tumor suppressive function of HIF2α.

The online version of this article includes the following figure supplement(s) for figure 9:

**Figure supplement 1.** ISGF3 subunits correlate positively with immune cell markers and negatively with cancer cell markers in the TCGA dataset.
**Figure supplement 2.** STAT2 and IRF9 protein levels are significantly higher in immune cells than that in kidney cells.
**Figure supplement 3.** IHC staining of ccRCC tumors with BAP1, IRF9 or STAT2 antibodies.

despite fully activated HIF. Only after combination with *Pbrm1* or *Bap1* conditional knockout can *Vhl* loss lead to kidney tumors (*Nargund et al., 2017*; *Gu et al., 2017*). Thus HIF could activate ISGF3, which constitutes a negative feedback loop in ccRCC. Its inhibitory effect on tumor growth can be relieved by the loss of PBRM1, KDM5C, SETD2 or BAP1 to promote robust tumor growth (*Figure 9E*).

ISGF3 is traditionally activated by interferon stimulation through phosphorylation of STAT1 and STAT2 (*Borden et al., 2007*). However, days after the initial activation, the posttranslational modifications (PTMs) disappear, yet ISGF3 can still maintain its activity as unphosphorylated ISGF (U-ISGF3) with greatly enhanced protein levels (*Cheon et al., 2013*). U-ISGF3 activates fewer targets than phosphorylated ISGF3, but it still potently antagonizes viral infection and DNA damage. In

oncology, ISGF3 is known to be a predictive biomarker for chemotherapy and radiation in breast cancer (*Weichselbaum et al., 2008*), and it is also associated with drug-resistance to immune checkpoint blockade (*Benci et al., 2016*). Recently, immune checkpoint inhibitors, specifically those that target Programmed Death 1 (PD-1), Programmed Death Ligand 1 (PD-L1), and Cytotoxic T-Lymphocyte Associated 4 (CTLA-4) have gained traction in metastatic RCC (mRCC). The FDA has granted priority review for ipilimumab/nivolumab in intermediate and poor-risk mRCC (*Abernathy, 2017*), with RCC experts declaring this combination the new standard of care in the treatment of mRCC (*Chustecka, 2017*; *Rini, 2017*). Loss-of-function mutations in *PBRM1* were discovered to confer clinical benefit to ccRCC patients treated with anti-PD-1 alone or in combination with anti-CTLA-4 therapies (*Miao et al., 2018*). In addition, *PBRM1* mutations were found to enhance T-cell-mediated killing in an unbiased, high-throughput screen in a melanoma model (*Pan et al., 2018*). As *PBRM1* and other ccRCC tumor suppressor genes (TSGs) converge on ISGF3, it will be very interesting to examine whether ISGF3 status is a predictive biomarker for response to immunotherapy. The fact that *PBRM1* mutations, but not those of *BAP1*, *SETD2*, and *KDM5C*, are a predictive biomarker for response to immunotherapy is likely because *PBRM1* and *BAP1* mutations are truncal (root change) in many cases while the mutations of the other two are subclonal (*Gerlinger et al., 2012*; *Gerlinger et al., 2014*). *BAP1* mutations might also elicit an additional change that cancels their immunotherapy-sensitizing effect.

ccRCC is known to harbor rampant intra-tumoral heterogeneity (*Gerlinger et al., 2012*; *Gerlinger et al., 2014*; *Jiang et al., 2016*; *Jiang et al., 2017*). Inactivating mutations in *PBRM1*, *SETD2*, *KDM5C* and *BAP1* tend to be focal in the tumors, so therapies that are designed to exploit the weakness induced by mutation of an individual gene could be problematic since it cannot treat the remaining cancer cells free of that mutation. Since the suppression of PBRM1, KDM5C, SETD2 or BAP1 all lead to the reduction of ISGF3, a potent tumor suppressor, it is possible that any effective therapeutic intervention that restores ISGF3 function could benefit ccRCC patients with a wide variety of mutations.

*Pbrm1* loss in mice and *PBRM1*-/- vs PBRM1 re-expressed human A704 cells showed that PBRM1 deficiency was linked to the activation of the JAK/STAT3 pathway (*Nargund et al., 2017*; *Miao et al., 2018*). STAT3 and ISGF3 are two different, often times opposing, pathways: ISGF3 is activated by type I IFN (*Cheon et al., 2013*), while STAT3 is activated by the IL6 family, oncogenes, and growth factors (*Heinrich et al., 1998*). They have very different gene targets. STAT3 is often considered an oncogene (*Yuan et al., 2016*), while STAT1 is usually considered a TSG (*Stephanou and Latchman, 2003*), and thus they appear to play opposing biological roles in tumorigenesis (*Avalle et al., 2012*). In human T lymphoma cells or mouse cells, loss of STAT3 leads to prolonged activation of STAT1 (*Regis et al., 2008*; *Costa-Pereira et al., 2002*; *Maritano et al., 2004*). In addition, STAT3 negatively regulates the type I IFN signaling pathway by inhibiting expression of *IRF7*, *IRF9*, *STAT1* in diffuse large B cell lymphoma (*Lu et al., 2018*). Thus STAT3 and ISGF3 are distinct and activated STAT3 might suppress ISGF3 in ccRCC cells, and this deserves further investigation.

In a recent publication, SETD2 was shown to methylate STAT1 to promote ISGF3 function (*Chen et al., 2017*). However, this requirement of SETD2 is not observed unless IFN is added to activate ISGF. In our system, no exogenous IFN was present, and HIF activation seemed to be sufficient to increase the protein levels of STAT2 and IRF9, thus it might activate U-ISGF3 (*Figure 5—figure supplement 1*). In 786-O cells with or without PBRM1 or KDM5C suppression, the mRNA levels of STAT1, STAT2 and IRF9 mirror that of the proteins (data not shown), suggesting that the regulation of U-ISFG3 occurred at the transcriptional level. Interestingly, BAP1 loss reduced the protein levels of STAT2 and IRF9 independent of PBRM1 status, since RCC4 cells express no endogenous PBRM1 (*Figure 6C*).

Why ISGF3 is a potent tumor suppressor in ccRCC tumors is currently not well understood. One possibility is that interferon stimulated genes (ISG) have anti-proliferative and pro-apoptotic function in ccRCC cancer cells. ISGs activated by IFN do have such activity to curb the spread of virus in the infected cells (*Borden et al., 2007*), and the U-ISGF3 was thought to be weaker in this regard with a much smaller target pool (*Cheon et al., 2013*). However, the U-ISGF3 may still possess sufficient anti-proliferative/pro-apoptotic activity to impair the fitness of the cancer cells. Our current evidence suggests that this does not apply in our model. Alternatively, U-ISGF3 may change how the cancer cells interact with the immune system. The nude mice used in our study do not have adaptive

immune system but the innate immune response remains intact. The heightened U-ISGF3 may attract more innate immune cells such as macrophages and NK cells to attack the tumors and curb tumor growth, as a previous report showed that over-expression of IFN-β recruited macrophages which suppressed tumor growth and metastasis in nude mice (*Zhang et al., 2002*). However, further studies to closely monitor the cancer and host cells are needed to provide a definitive answer to this question. It will be important to identify the mechanism of how ISGF3 curbs tumor growth in ccRCC with human tumor samples and immune-competent mouse models so we can exploit this pathway to treat ccRCC patients.

With our current preliminary data we think that PBRM1 and KDM5C regulate ISGF3 subunits at transcriptional level, while SETD2 and BAP1 regulate ISGF3 at post-translational modification level. They might use quite different mechanisms to achieve the same goal: maintaining the ISGF3 activity, so loss of any of them could promote tumor development. Further investigation into these mechanisms is needed.

It is highly surprising that so many major TSGs in ccRCC share the same target in ISGF3. Undoubtedly each TSG has its own unique tumor suppressing functions, as each carries different prognoses for the patients. *KDM5C* mutations were reported to occur in the cells with *PBRM1* mutations (*Hsieh et al., 2017*). It is possible that their unique tumor suppressor functions are synergistic so their mutations are selected in the same cells during tumor development. However, the fact that they all converge on ISGF3 warrants further investigation into how this pathway impacts drug response, patient survival, animal modeling and drug development.

# Materials and methods

## Key resources table

| Reagent type (species) or resource | Designation | Source or reference | Identifiers | Additional information |
|---|---|---|---|---|
| Cell line (Human) | 786-O | purchased from ATCC | ATCC CRL-1932 | |
| Cell line (Human) | A498 | purchased from ATCC | ATCC CRL-7908 | |
| Cell line (Human) | Ren-01 | Daniel Lindner at Cleveland Clinic | | |
| Cell line (Human) | Ren-02 | Daniel Lindner at Cleveland Clinic | | |
| Cell line (Human) | RCC4 | William Kaelin Jr. at Dana-Farber Cancer Institute | | |
| Cell line (Human) | Caki-1 | William Kaelin Jr. at Dana-Farber Cancer Institute | | |
| Cell line (Human) | HEK293T | William Kaelin Jr. at Dana-Farber Cancer Institute | | |
| Cell line (Human) | UMRC2 | Qing Zhang at UNC Chapel Hill | | |
| Cell line (Human) | UMRC6 | Qing Zhang at UNC Chapel Hill | | |
| Antibody | BAP1 (mouse monoclonal) | Santa Cruz Biotech, sc-28383 | | (1:200 for western blot, 1:50 for IHC) |
| Antibody | STAT2 (rabbit polyclonal) | Bethyl A303-512A | | (1:1,000 for western blot, 1:25 for IHC) |
| Antibody | IRF9 (rabbit polyclonal) | Sigma, HPA001862 | | (1:2,000 for western blot, 1:50 for IHC) |
| Recombinant DNA reagent | Flag-PBRM1 (human) | This paper | | Available upon request. |
| Recombinant DNA reagent | Flag-KDM5C (human) | This paper | | Available upon request. |

*Continued on next page*

Continued

| Reagent type (species) or resource | Designation | Source or reference | Identifiers | Additional information |
|---|---|---|---|---|
| Other | Ren-02 cells with BAP1 knocked out | This paper | | BAP1 gene was knocked out with the CrRISPR-Cas9 technique. Detailed description was listed in the Materials and methods. |

## Gene expression analysis

Genome-wide gene transcription was profiled using Illumina HumanHT-12_V4 Expression Bead Chip. The raw gene expression data were preprocessed in Genome Studio (https://www.illumina.com/techniques/microarrays/array-data-analysis-experimental-design/genomestudio.html), and further processed using R limma package (Ritchie et al., 2015) for force-positive background correction, $\log_2$ transformation and quantile normalization. Data quality control was performed by selecting present probes using the detection threshold of 0.05. Outlier samples were examined using hierarchical clustering and no samples were removed. The average value across the probesets of an annotated gene was used as the gene-level expression and the data matrix was exported for downstream analysis. Differential gene expression analyses were performed using limma package controlling for batch effects in a linear model (Phipson et al., 2016). Gene set enrichment was performed using a preranked GSEA algorithm (Subramanian et al., 2005) based on $\log_2$ fold change. GO analyses were performed using the DAVID (Huang et al., 2009) online tools. For the GO analyses, Venn diagrams, and heatmaps, the cutoff of adjusted p-value less than 0.05 and fold change greater than two were applied to select genes. The statistical significance of the overlaps in the Venn diagrams was calculated based on hypergeometric distribution function – phyper implemented in R (https://www.r-project.org/). The gene expression values used in the heatmaps were preprocessed by ComBat (Johnson et al., 2007) to remove batch effects, and then centered by the mean values of each row. Source file: GEO accession GSE108229.

## Analysis of TCGA dataset

Gene expression for the TCGA dataset, Kidney renal cell carcinoma (KIRC), were obtained through the 'cgdsr' R package (Jacobsen and cBioPortal Questions, 2018). RSEM expression data were transformed into log2(RSEM +1). Transformed data were used for Pearson correlation and plotted with the 'ggplot2' R package (Wickham, 2016). Scripts available upon request.

## Analysis of TMA

Association between different IHC targets were determined using Spearman correlation. Kaplan-Meier survival analyses were performed and plotted using 'survival' and 'survminer' R packages (Kassambara et al., 2018; Therneau and Lumley, 2015; Therneau and Patricia, 2000). Statistical significance of survival curves was calculated using the log-rank test.

## Mutagenesis, subcloning and primers for RT-PCR

Mutagenesis was performed via the QuikChange method using phusion DNA polymerase (Thermo-Fisher). Primers were designed using Agilent's QuikChange Primer Design program. The sequences are as follows: BAP1-N78S-F: CTGGTGGGCAAAGAACATGTTACTCACAATATCATCATCAATCAC, BAP1-N78S-R: GTGATTGATGATGATATTGTGAGTAACATGTTCTTTGCCCACCAG, BAP1-G185R-F: GACCTTCAGCCTATCCAGCTCAAAGAGCCG, BAP1-G185R-R: CGGCTCTTTGAGCTGGATAGGCTGAAGGTC. A temperature gradient was performed to ensure primer annealing during PCR cycling. Following DpnI digestion, DNA was transformed into XL10 Gold Ultracompetent Cells (Agilent). Plasmids were sequenced to confirm the point mutations.

Full-length PBRM1 was subcloned into p3xFlag-CMV10 with primers PBRM1-F: CCCAAGCTTATGGGTTCCAAGAGAAGAAG and PBRM1-R: GGAATTCTTAAACATTTTCTAGGTTGT. For 1–1297 and 1–1184 PBRM1 constructs, the following primers were used to introduce nonsense mutants through mutagenesis:

1297 (3892C-T): 3892 F: CCAATTGTTCCTTAGAAGGAGCCAT; 3892 R: ATGGCTCCTTCTAAG-GAACAATTGG

1184 (3553C-T): 3553 F: AAAGTATGGGTTTGAGATGGAGCTG; 3553 R: CAGCTCCATCTCAAACCCATACTTT

## Primers for real-time PCR

| Primer name | Primer sequence |
| --- | --- |
| GAPDH-F | TGCACCACCAACTGCTTAGC |
| GAPDH-R | GGCATGGACTGTGGTCATGAG |
| OAS1-F | CATCTGTGGGTTCCTGAAGG |
| OAS1-R | GAGAGGACTGAGGAAGACAAC |
| OAS2-F | CAGTCCTGGTGAGTTTGCAGT |
| OAS2-R | ACAGCGAGGGTAAATCCTTGA |
| IFI44-L | TTCGATGCGAAGATTCACTG |
| IFI44-R | CCCTTGGAAAACAGACCTCA |
| IFI44L-F | GAGCACAGAAATAGGCTTCTAGC |
| IFI44L-R | TGGTATCAGACCCCACTACGG |
| IFIT1-F | CTGAATGCAGCTCACCTCTG |
| IFIT1-R | GGATGGAATTGCCTGCTAGA |
| ISG20-F | GTCACCCCTCAGCACATG |
| ISG20-R | AGATTGTGTAGCCGCTCATG |
| GAS6-F | AGACTATCACTCCACGAAG |
| GAS6-R | TCGCAGACCTTGATCTCC |
| STMN3-F | CAGCACCATTTCCGCCTA |
| STMN3-R | CCTCCATGTCCCCGTA |
| ARPC4-F | TCAGTCAATGCCCGTG |
| ARPC4-R | CTCCAGGACGCCTTCG |
| IL6-F | CCAGCTATGAACTCCTTCTC |
| IL6-R | GCTTGTTCCTCACATCTCTC |
| IL8-F | ATCGCTTCCTCTCGCAACAA |
| IL8-R | CTTCTACTGGTTCAGCAGCCATCT |
| CCL5-F | TCTGCGCTCCTGCATCTG |
| CCL5-R | GGGCAATGTAGGCAAAGCA |
| SETD2-F: | ATCGAGAGAGGACGCGCTATT |
| SETD2-R: | AGGTACGCCTTGAGTATGTCTT |

## Cell culture and viral infection

Kidney cancer cell lines 786-O and A498 were purchased from ATCC. Ren-02, RCC4 and HEK293T were described previously (*Niu et al., 2012*; *Negrotto et al., 2011*). UMRC2 and UMRC6 cells were a gift from Dr. Qing Zhang from University of North Carolina at Chapel Hill. All cell lines were maintained in 37℃ incubator with 5% $CO_2$ in glutamine-containing DMEM medium supplemented with 10% fetal bovine serum (FBS) and 1% penicillin and streptomycin. Mycoplasma contamination is tested with a PCR-based kit. The cells will be treated with antibiotics to eradicate the mycoplasma if it is detected. The cell proliferation rate was measured with a WST-1 kit from Sigma Aldrich according to the manufacturer's instruction.

The packaging of shRNA viral particles, the infection and selection, and the expression of shRNA resistant plasmid were the same as described before (*Kondo et al., 2002*).

## The generation of Ren-02 BAP1 knockout (KO) clone

Ren-02 BAP1 knockout (KO) cells were generated using an inducible CRISPR-Cas9 system previously described (Cao et al., 2016). Briefly, HEK293T cells were transfected with 2 µg pLenti-iCas9-Neo, 2µgΔR8.9 packaging vector, and 200 ng VSV-G envelope vector using lipofectamine 2000. Lentiviral supernatants were filtered and used to infect Ren-02 cells with polybrene, and infected cells were selected with 1.0 mg/ml G418. The cells were then treated with 1.0 µg/ml doxycycline to induce Cas9 expression, and sorted by GFP expression using flow cytometry. Short guide RNA (sgRNA) sequences were designed to target BAP1 exon five using Optimized Crispr Design (crispr.mit.edu), using the following oligonucleotides: BAP1-exon5-F: CACCGACCCACCCTGAGTCGCATGA; BAP1-exon5-R: AAACTCATGCGACTCAGGGTGGGTC. The sgRNAs were cloned into pLentiGuide-Puro as previously described (Sanjana et al., 2014; Shalem et al., 2014). The pLentiGuide-Puro-sgRNA plasmids were transfected into HEK293T cells to generate lentivirus, which was used to infect Ren-02 iCas9 cells. Following selection with 2.0 µg/ml puromycin, the Ren-02 iCas9-sgRNA cells were treated with doxycycline, diluted, and screened to identify individual clones with BAP1 KO.

## Western blot and immunoprecipitation

Cells were washed in PBS buffer, then lysed with EBC buffer (50 mM Tris, pH 8.0; 120 mM NaCl; 0.5% NP-40) supplemented with a protease inhibitor cocktail. In some cases 1% SDS solution was added to the cells in the plates instead of EBC buffer, then the solution was sonicated at 20 A for 3 × 15 s pulses and mixed with sample buffer. The same total amount of total protein was loaded and resolved by SDS–PAGE and analyzed by immunoblotting. The blots were developed with Super Signal Pico substrate (Pierce Biotechnology, Rockford, IL) or Immobilon Western substrate (Millipore, Billerica, MA). Immunoprecipitations were performed with EBC lysates as previously described (Niu et al., 2012).

The antibodies used in this manuscript are: HA (Covance MMS-101P-1000 HA.11 (16B12 for IB, Santa Cruz Biotech sc-805 for IP), Flag (Sigma F3165-1mg (M2)), KDM5C (Bethyl A301-034A and A301-035A), BAP1 (Santa Cruz Biotech sc-28383), γ-Actin (Santa Cruz Biotech sc-8432), PBRM1 (Bethyl A301-591A and A301-590A), STAT1 (Cell signaling 9172S), phospho-STAT1 (Cell signaling 9171L), STAT2 (Bethyl A303-512A-T), phospho-STAT2 (Cell signaling 88410), IRF9 (Cell signaling 76684S), MX1 (R and D AF7946), PLSCR1 (Santa Cruz Biotech sc-59645), Vinculin (Santa Cruz Biotech sc-73614), GAPDH (Santa Cruz Biotech sc-59540), HIF2α (Novus NB100-122), OAS1 (Cell signaling 14498S).

## Short hairpin RNAs (shRNAs)

All the shRNA constructs except for SCR were obtained from Sigma (St Louis, MO). The sequences were: SCR: GCGCGCUUUGUAGGAUUCGTT; HIF2a-17: CGACCTGAAGATTGAAGTGAT; HIF2a-20: CCATGAGGAGATTCGTGAGAA; PBRM1-94: CCGGAGTCTTTGATCTACAAA; PBRM1-890: CCGGAATGCCAGGCACTATA; BAP1-71: CGTCCGTGATTGATGATGATA, BAP1-74: CCACAAC TACGATGAGTTCAT; SETD2-30: CCTGAAGAATGATGAGATAAT; SETD2-32: GCCCTATGACTCTC TTGGTTA; IRF9-69: GCCATACTCCACAGAATCTTA.

## Nude mice xenograft analysis

All animal experiments were conducted in accordance with protocol 01462-935A approved by the IACUC of Thomas Jefferson University and protocol 2015–11286 approved by the IACUC of Yale University. The subcutaneous nude-mice xenograft assay was performed as described (Kondo et al., 2002; Yan et al., 2007). For each cell line, $10^7$ cells were injected subcutaneously into the flanks of immunocompromised male nu/nu nude mice of four weeks old purchased from Taconic or Jackson Laboratory. All mice were sacrificed by $CO_2$ inhalation 8 to 10 weeks after injection of cells, and tumors were excised and weighed. Results are reported as mean ±se. of the mean. Nine to ten pairs of tumors were compared. Results were statistically evaluated with Mann–Whitney U statistic analysis.

## Immunohistochemistry staining and image analysis

4 µm paraffin slides were deparaffinized in Shandon Varistain Gemini ES Autostainer. Antigen retrieval was performed with DAKO PTLink using Citrate Buffer (pH6.0) at 98 °C for a total time of

20 min. Primary immunostaining was performed using antibodies against Ki67 (abcam, Cat#: ab16667, 1:200), Cleaved Caspase 3 (Cell-Signaling, cat#:9661, 1:500) and CD45 (BD Pharmingen, Cat#: 550539, 1:200), BAP1 (Santa Cruz Biotech, sc-28383, 1:50), STAT2 (Bethyl A303-512A, 1:25), IRF9 (Sigma, HPA001862, 1:50). Antibodies were incubated at room temperature for 30 min. Biotinylated anti-Rabbit or anti-Rat (Vector Laboratories, cat#: BA-1000 and BA-4001) secondary antibodies and ABC-HRP complexes (Vector Laboratories, Cat#: PK6100) were applied following the primary antibodies with 30 min incubation of each reagent at room temperature. Three TBST washes were performed between each step above. The signals were visualized using DAB substrate (DAKO, Cat#: K3468). Slides were then washed with DI water and further processed with Hematoxylin counter stain, dehydration and clearing in Shandon Varistain Gemini ES Autostainer and coverslipped with Permount Mounting Medium.

## Acknowledgement

We are very grateful to Dr. Robert Silverman at Cleveland Clinic for the suggestion that ISGF3 might be responsible for the shared target genes. We thank Dr. Wei Xu at University of Wisconsin for providing the p-LNCX expression vector. We also acknowledge Dr. Qing Zhang at the University of North Carolina at Chapel Hill for providing the UMRC2 and UMRC6 cell lines. Research reported in this publication utilized the Translational Pathology Shared Resource at Sidney Kimmel Cancer Center at Jefferson Health and was supported by the National Cancer Institute of the National Institutes of Health under Award Number **P30CA056036**. The content is solely the responsibility of the authors and does not necessarily represent the official views of the NIH.

## Additional information

### Funding

| Funder | Grant reference number | Author |
| --- | --- | --- |
| National Cancer Institute | R01 CA155015 | Haifeng Yang |
| National Cancer Institute | P30CA056036 | Haifeng Yang |
| Department of Defense | W81XWH-16-1-0326 | Qin Yan |

The funders had no role in study design, data collection and interpretation, or the decision to submit the work for publication.

### Author contributions

Lili Liao, Conceptualization, Resources, Supervision, Funding acquisition, Investigation, Writing—original draft, Project administration, Writing—review and editing; Zongzhi Z Liu, Weijia Cai, Validation, Investigation, Writing—review and editing; Lauren Langbein, Data curation, Formal analysis, Visualization; Eun-Ah Cho, Jie Na, Wei Jiang, Validation, Investigation; Xiaohua Niu, Formal analysis, Investigation; Zhijiu Zhong, Formal analysis, Investigation, Visualization; Wesley L Cai, Validation, Investigation, Methodology; Geetha Jagannathan, Formal analysis, Writing—review and editing; Essel Dulaimi, Formal analysis; Joseph R Testa, Robert G Uzzo, Yuxin Wang, George R Stark, Jianxin Sun, Resources; Stephen Peiper, Yaomin Xu, Resources, Writing—review and editing; Qin Yan, Data curation, Formal analysis, Supervision, Methodology, Writing—review and editing, Funding Acquisition; Haifeng Yang, Resources, Data curation, Software, Formal analysis, Supervision, Funding acquisition, Methodology, Writing—review and editing

### Author ORCIDs

Lauren Langbein (ID) http://orcid.org/0000-0002-3007-5287
Yaomin Xu (ID) http://orcid.org/0000-0002-3752-4006
Qin Yan (ID) https://orcid.org/0000-0003-4077-453X
Haifeng Yang (ID) http://orcid.org/0000-0002-0892-9055

## Ethics

Animal experimentation: All animal experiments were conducted in accordance with protocol 01462-935A approved by the IACUC of Thomas Jefferson University and protocol 2015-11286 approved by the IACUC of Yale University.

## Decision letter and Author response

Decision letter https://doi.org/10.7554/eLife.37925.sa1
Author response https://doi.org/10.7554/eLife.37925.sa2

## Additional files

### Supplementary files

• Transparent reporting form

### Data availability

Microarray data have been deposited inn GEO under the accession code GSE108229.

The following dataset was generated:

| Author(s) | Year | Dataset title | Dataset URL | Database and Identifier |
|---|---|---|---|---|
| Liao L, Liu Z, Na J, Niu X, Xu Y, Yan Q, Yang H | 2018 | Microarray analysis of gene expression after suppression of PBRM1 or KDM5C in 786-O VHL+/+ or VHL-/- cells | https://www.ncbi.nlm.nih.gov/geo/query/acc.cgi?acc=GSE108229 | NCBI Gene Expression Omnibus, GSE108229 |

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
