## [Decision Letter]

Thank you for submitting your article "Multiple Tumor Suppressors Regulate a HIF-dependent Negative Feedback Loop through ISGF3 in Kidney Cancer" for consideration by *eLife*. Your article has been reviewed by three peer reviewers, including Irwin Davidson as the Reviewing Editor and Reviewer #1, and the evaluation has been overseen by Jeffrey Settleman as the Senior Editor. The following individual involved in review of your submission has agreed to reveal their identity: James Hsieh (Reviewer #3).

The reviewers have discussed the reviews with one another and the Reviewing Editor has drafted this decision to help you prepare a revised submission.

Summary:

This paper describes the role of secondary mutations in ccRCC cells affecting PBRM1, KDM5C, BAP1 and SETD2; multiple tumor suppressors frequently mutated in ccRCC that converge on a HIF-dependent negative feedback loop through ISGF3. ShRNA knockdown of these genes in ccRCC cells in vitro, taken as a surrogate for the normally observed inactivating mutations seen in human tumours, lowers the expression of the STAT1, STAT2 and/or IRF9 proteins that together form the ISGF3 transcription factor. The study shows that ISGF3 is a potent tumour suppressor that does not act principally through direct regulation of cell proliferation, but modulates the immune-infiltrate of the tumours in some undescribed way to affect tumour growth.

Essential revisions:

All three reviewers thought that this was an interesting study with novel and relevant conclusions, but they found a number of important issues that require serious attention in any revised version.

The experiments performed in this study used established ccRCC cell lines. To verify that the observations are relevant for human ccRCC, the authors should use TCGA ccRCC datasets to investigate expression of ISGF3 in human ccRCC and its prognostic impact. They should use the TCGA dataset to test the hypothesis that tumors with no PBRM1, BAP1, SETD2 and KDM5C have differential expression of ISGF3 components. It is also essential to show by immunostaining that ISGF3 subunits are really down-regulated in sections form human ccRCC where the above genes are inactivated. It is critical that the mechanisms described here using model cells lines and xenograft tumours are verified in one way or another in panels of human ccRCC.

The authors report that KDM5C and PBRM1 physically interact. Figure 4 shows that expression of FLAG-PBRM1 can precipitate endogenous KDM5C, but the opposite has not been shown by transfecting FLAG-KDM5C to pull down endogenous PBRM1. This should be further investigated. PBRM1 is a well characterized subunit of the PBAF complex. The results presented here suggest that KDM5C associates with PBAF through interactions with PBRM1. This is a potentially important finding, but should be confirmed. The authors should investigate whether tagged PBRM1 co-precipitates other subunits of the PBAF complex and they should verify whether tagged KDM5C can precipitate additional subunits of the PBAF complex. In particular, they should exclude the possibility that these proteins interact when overexpressed because PBRM1 does not find its normal interaction partners in PBAF. The interactions between endogenous proteins also need to be demonstrated in ccRCC cell lines. The authors should also comment on the fact that KDM5C mutations tend to occur under the PBRM1 mutated background (PMID 27751729). How can these observations be rationalized?

The authors show that knockdown of IRF9 leads to much larger xenograft tumours than control cells. Did they do similar experiments with STAT1 or STAT2? If not what is the explanation? If there is no caveat, they should also show that silencing of either STAT1 or STAT2 has the same tumour-promoting effect. Stable silencing of IRF9 clearly affects its own expression, but does it result in lower STAT1 and STAT2 expression? Can the authors demonstrate that down-regulated IRF9 expression is maintained during tumour growth and in the endpoint tumours?

The authors show increased immune-infiltrate in the IRF9 knockdown cells, but only a mild increase in KI67 staining. If there is no strong increase in proliferation, why are the tumours so much bigger? Are the differences due to ISGF3 affecting apoptosis and/or senescence? This should be clarified. Similarly, overexpression of IRF9 and STAT2 strongly inhibits tumour growth, is this accounted for by decreased proliferation, or increased apoptosis/senescence?

Some important controls are missing: Figure 5F do not show knockdown of KDM5C and PBRM1 by western blot; Figure 6A and 6C do not show knockdown of SETD2 by western blot (RT-PCR shows minimal knock-down); Figure 6A, 6B and 6C have overexposed GAPDH signal, Figure 6 has no letter "A" in its tagline.

We suggest to repeat the experiment in Figure 5—figure supplement 4 in KDM5C knock-down.

---

## [Author Response]

Essential revisions:All three reviewers thought that this was an interesting study with novel and relevant conclusions, but they found a number of important issues that require serious attention in any revised version.The experiments performed in this study used established ccRCC cell lines. To verify that the observations are relevant for human ccRCC, the authors should use TCGA ccRCC datasets to investigate expression of ISGF3 in human ccRCC and its prognostic impact. They should use the TCGA dataset to test the hypothesis that tumors with no PBRM1, BAP1, SETD2 and KDM5C have differential expression of ISGF3 components. It is also essential to show by immunostaining that ISGF3 subunits are really down-regulated in sections form human ccRCC where the above genes are inactivated. It is critical that the mechanisms described here using model cells lines and xenograft tumours are verified in one way or another in panels of human ccRCC.

We thank the reviewers for the thoughtful comments and we agree that our mechanism need to be verified in human ccRCC panel. We first conducted analysis of TCGA dataset as suggested by the reviewers. However, in many cases, we did not find consistent correlations between mRNA levels of BAP1, PBRM1, KDM5C, SETD2 and ISGF3 components in the whole tumors (Author response image 1). There were a few positive correlations and one case of negative correlation. One possibility for these results is that the mRNA expression of ISGF3 components detected in TCGA is not primarily coming from cancer cells, but from other stromal cells which can distort the results. To investigate this, we analyzed the correlations between well-known immune and tumor cell markers (CD45 or PTPRC, CD3, CD8, etc. and CDH1, EPCAM) with ISGF3 subunits within the TCGA dataset. The graphs in Figure 9—figure supplement 1 clearly indicate that the mRNA expression of ISGF3 subunits in the TCGA dataset comes predominantly from immune cells as opposed to tumor cells (surrogated by EPCAM and E cadherin).

**Author response image 1. respfig1:** The correlations between ISGF3 subunits and PBRM1, KDM5C, BAP1 or SEDT2 in the TCGA dataset. Correlations between IRF9, STAT1 or STAT2 with indicated genes within the TCGA dataset. N = 533.

We also sought to confirm this with immunohistochemistry (IHC). We found that immune cells had much higher protein expression of ISGF3 subunits STAT2 than the normal or cancerous human kidney cells. On the same slide, the immune cells in lymph node, spleen and lung tissues were stained strongly positive with antibody against STAT2, but the normal and cancerous kidney tissue stained negative or very weakly (Figure 9—figure supplement 2A). Interestingly, in the lung tissue, the strong staining of STAT2 in the immune cells did not seem to significantly increase the staining intensity of the surrounding cells. Similar observation was made with IRF9 IHC (Figure 9—figure supplement 2B). Thus immune cells have much higher expression of ISGF3 subunits STAT2 and IRF9 than that in kidney cancer cells, and this could disrupt the detection of ISGF3 mRNA expression in the cancer cells. Since our hypothesis is that BAP1, PBRM1, KDM5C, SETD2 are critical to maintain the expression and/or function ofISGF3 subunits in the ccRCC cancer cells, we concluded that mRNA levels in the bulk tumors from the TCGA dataset are not suitable for our validation experiments.

To validate our hypothesis, we decided to examine the ISGF3 protein levels in ccRCC cancer cells with IHC and correlate their protein levels with the protein levels of BAP1, PBRM1, and SETD2. We have previously examined the protein levels of PBRM1 and SETD2 in a ccRCC tissue microarray (TMA) generated by our colleagues at Fox Chase Cancer Center in our previous publication (Jiang et al., 2016). This TMA includes 40 cases per tumor stage, and for each tumor 4 foci was chosen except for one case (total 638 foci). We probed the same TMA with antibodies against BAP1, IRF9 and STAT2. The staining ofBAP1 or IRF9 mostly occurs in the nucleus, while STAT2 staining is found in both cytoplasm and nucleus (Figure 9A). In many cases the protein expression levels of PBRM1, SETD2 or BAP1 correlated with that of STAT2 or IRF9 (Figure 9A). After scoring the samples and performing statistical analysis, the Spearman association analysis revealed that nuclear IRF9 and cytoplasmic STAT2 both significantly correlated with the expression of PBRM1, SETD2, or BAP1 (Figure 9B). The correlations were mostly statistically significant when all the foci were considered. In individual tumor stage or grade, the associations were mostly statistically significant as well. Interestingly, the loss of nuclear staining of IRF9 was associated with worse patient survival, consistent with a tumor suppressor role of ISGF3 (Figure 9C). The loss of cytoplasmic STAT2staining was not associated with worse patient survival (Figure 9D), suggesting that other ISGF3 components are the more dominant players in the human ccRCC. Consistent with this idea, the change of IRF9 protein in the TMA (Figure 9B) and in the cultured cells (Figures 5 and 6) tracked better with the changes of PBRM1, SETD2 or BAP1. Taken together, our data suggests that the links between PBRM1, SETD2 or BAP1 and ISGF3 are preserved in human ccRCC tumors. Wesley L. Cai, Geetha Jagannathan, Essel Dulaimi, Joseph R. Testa and Robert Uzzo contributed to this new analysis and are therefore added as co-authors.

The authors report that KDM5C and PBRM1 physically interact. Figure 4 shows that expression of FLAG-PBRM1 can precipitate endogenous KDM5C, but the opposite has not been shown by transfecting FLAG-KDM5C to pull down endogenous PBRM1. This should be further investigated. PBRM1 is a well characterized subunit of the PBAF complex. The results presented here suggest that KDM5C associates with PBAF through interactions with PBRM1. This is a potentially important finding, but should be confirmed. The authors should investigate whether tagged PBRM1 co-precipitates other subunits of the PBAF complex and they should verify whether tagged KDM5C can precipitate additional subunits of the PBAF complex. In particular, they should exclude the possibility that these proteins interact when overexpressed because PBRM1 does not find its normal interaction partners in PBAF. The interactions between endogenous proteins also need to be demonstrated in ccRCC cell lines. The authors should also comment on the fact that KDM5C mutations tend to occur under the PBRM1 mutated background (PMID 27751729). How can these observations be rationalized?

We thank the reviewers for your suggestion. We performed a series of experiments as the reviewers suggested. Flag-KDM5C was able to pull down endogenous PBRM1 along with BRG1 and BAF170, the catalytic and structural subunits of PBAF complex respectively (Figure 4C). Interestingly, it did not significantly co-IP BRD7 or BAF57, unlike that of Flag-PBRM1 (Figure 4B), suggesting that Flag-KDM5C pulled down a partial PBAF complex. Moreover, the endogenous PBRM1 also pulled down the endogenous KDM5C in human embryonic kidney (HEK) 293T cells along with the other PBAF subunits (Figure 4D), suggesting that they do exist in one complex in kidney cells and it is not an artifact due to over-expression. The exact mechanism of this interaction and it impact on the functions of PBRM1 and KDM5C await further investigation.

As to the co-occurrence of mutations in PBRM1 and KDM5C, we stated that “undoubtedly each TSG has its own unique tumor suppressing functions so each carries different prognosis for the patients” in our Discussion. It is possible that the unique tumor suppressor functions of PBRM1 and KDM5C are synergistic so their mutations are selected in the same cells during tumor development. This point has been added to the Discussion.

The authors show that knockdown of IRF9 leads to much larger xenograft tumours than control cells. Did they do similar experiments with STAT1 or STAT2? If not what is the explanation? If there is no caveat, they should also show that silencing of either STAT1 or STAT2 has the same tumour-promoting effect. Stable silencing of IRF9 clearly affects its own expression, but does it result in lower STAT1 and STAT2 expression? Can the authors demonstrate that down-regulated IRF9 expression is maintained during tumour growth and in the endpoint tumours?

This is a great suggestion and we agree that it is important to examine whether the suppression of STAT1 or STAT2 would boost tumor growth. We stably suppressed the expression of either STAT1 or STAT2 in 786-O cells then performed xenograft analysis. In both cases, the tumors were significantly bigger than the ones made by the control cells (Figure 7—figure supplement 3). We also found that IRF9 suppression did not suppress the expression of other subunits of ISGF3 (Figure 7—figure supplement 2A), and IRF9 suppression was maintained in the endpoint tumors (Figure 7—figure supplement 2B). The tumors made with the control cells were very small even at the end point, so it was not practical to examine IRF9 suppression during tumor growth.

The authors show increased immune-infiltrate in the IRF9 knockdown cells, but only a mild increase in KI67 staining. If there is no strong increase in proliferation, why are the tumours so much bigger? Are the differences due to ISGF3 affecting apoptosis and/or senescence? This should be clarified. Similarly, overexpression of IRF9 and STAT2 strongly inhibits tumour growth, is this accounted for by decreased proliferation, or increased apoptosis/senescence?

Thank for your suggestion and we have conducted several experiments to address these. However, we do not have a definitive answer to why the tumors were much bigger after the suppression of ISGF3 subunits. The apoptosis and proliferation, measured by cleaved caspase 3 and Ki-67 respectively, did not show great difference (Figure 5—figure supplement 2). We were unable to measure senescence in the tumor samples. Overexpression of IRF9 and STAT2 strongly inhibited tumor growth, and our examination did not reveal any major difference in cell proliferation or apoptosis either, at least within the tumors. Thus it is possible that higher ISGF3 activity recruited host immune cells to kill the cancer cells, which cannot be detected by the assays described above. If the attrition mostly happened at the outer layer, the dead cells might dissociate form the tumor and cannot be observed. It is also possible that other forms of cell death such as necrosis or ferroptosis happened within the tumor that we failed to detect with our methods. The exact mechanism will require further investigation with more sensitive methods to monitor the cancer cells and host cells including the immune cells. In light of this, we modified our discussion to “A possible mechanism is that heightened U-ISGF3 may attract more innate immune cells such as macrophages and NK cells to attack the tumors and curb tumor growth, as a previous report showed that over-expression of IFN-β recruited macrophages which suppressed tumor growth and metastasis in nude mice (Zhang, Lu and Dong, 2002). However, further studies to closely monitor the cancer and host cells are needed to provide a definitive answer to this question. “

Some important controls are missing: Figure 5F do not show knockdown of KDM5C and PBRM1 by western blot; Figure 6A and 6C do not show knockdown of SETD2 by western blot (RT-PCR shows minimal knock-down); Figure 6A, 6B and 6C have overexposed GAPDH signal, Figure 6 has no letter "A" in its tagline.

Thanks for pointing out these issues. We added western blots of knockdown of KDM5C and PBRM1 to Figure 5F.

We were able to detect SETD2 knockdown by western blot in 6A but not in 6C (probably due to low expression of SETD2 protein in RCC4 cells), and it was added as Figure 6—figure supplement 1.

Lighter GAPDH exposures were used replace the GAPDH blots in Figure 6A-C. Similar number of cells was used to generate lysates. Letter A in Figure 6A was added.

We suggest to repeat the experiment in Figure 5—figure supplement 4 in KDM5C knock-down.

As suggested, we performed the experiment and found that similar to PBRM1, KDM5C does not affect phosphorylation of ISGF3. We added these results to Figure 5—figure supplement 4.